# Dark Experience for General Continual Learning: a Strong, Simple Baseline

**Pietro Buzzega**    **Matteo Boschini**    **Angelo Porrello**    **Davide Abati**    **Simone Calderara**

AImageLab - University of Modena and Reggio Emilia, Modena, Italy
`name.surname@unimore.it`

## Abstract

Continual Learning has inspired a plethora of approaches and evaluation settings; however, the majority of them overlooks the properties of a practical scenario, where the data stream cannot be shaped as a sequence of tasks and offline training is not viable. We work towards *General Continual Learning* (GCL), where task boundaries blur and the domain and class distributions shift either gradually or suddenly. We address it through mixing rehearsal with knowledge distillation and regularization; our simple baseline, *Dark Experience Replay*, matches the network's logits sampled throughout the optimization trajectory, thus promoting consistency with its past. By conducting an extensive analysis on both standard benchmarks and a novel GCL evaluation setting (MNIST-360), we show that such a seemingly simple baseline outperforms consolidated approaches and leverages limited resources. We further explore the generalization capabilities of our objective, showing its regularization being beneficial beyond mere performance. Code is available at https://github.com/aimagelab/mammoth.

## 1 Introduction

Practical applications of neural networks may require to go beyond the classical setting where all data are available at once: when new classes or tasks emerge, such models should acquire new knowledge on-the-fly, incorporating it with the current one. However, if the learning focuses on the current set of examples solely, a sudden performance deterioration occurs on the old data, referred to as catastrophic forgetting [29]. As a trivial workaround, one could store all incoming examples and re-train from scratch when needed, but this is impracticable in terms of required resources. Continual Learning (CL) methods aim at training a neural network from a stream of non i.i.d. samples, relieving catastrophic forgetting while limiting computational costs and memory footprint [32].

It is not always easy to have a clear picture of the merits of these works: due to subtle differences in the way methods are evaluated, many state-of-the-art approaches only stand out in the setting where they were originally conceived. Several recent papers [10, 11, 17, 39] address this issue and conduct a critical review of existing evaluation settings, leading to the formalization of three main experimental settings [17, 39]. By conducting an extensive comparison on them, we surprisingly observe that a simple Experience Replay baseline (*i.e.* interleaving old examples with ones from the current task) consistently outperforms cutting-edge methods in the considered settings.

Also, the majority of the compared methods are unsuited for real-world applications, where memory is bounded and tasks intertwine and overlap. Recently, [10] introduced a series of guidelines that CL methods should realize to be applicable in practice: i) **no task boundaries**: do not rely on boundaries between tasks during training; ii) **no test time oracle**: do not require task identifiers at inference time; iii) **constant memory**: have a bounded memory footprint throughout the entire training phase.

| Methods | PNN [35] | PackNet [28] | HAT [37] | ER [31, 33] | MER [33] | GSS [1] | GEM [27] | A-GEM [9] | HAL [8] | iCaRL [32] | FDR [4] | LwF [24] | SI [42] | oEWC [20] | **DER** (ours) | **DER++** (ours) |
|---|---|---|---|---|---|---|---|---|---|---|---|---|---|---|---|---|
| **Constant memory** | – | – | – | ✓ | ✓ | ✓ | ✓ | ✓ | ✓ | ✓ | ✓ | ✓ | ✓ | ✓ | ✓ | ✓ |
| **No task boundaries** | – | – | – | ✓ | ✓ | ✓ | – | ✓ | – | – | – | – | – | – | ✓ | ✓ |
| **No test time oracle** | – | – | – | ✓ | ✓ | ✓ | ✓ | ✓ | ✓ | ✓ | ✓ | – | ✓ | ✓ | ✓ | ✓ |

Table 1: Continual learning approaches and their compatibility with the General Continual Learning major requirements [10]. For an exhaustive discussion, please refer to supplementary materials.

These requirements outline the General Continual Learning (GCL), of which Continual Learning is a relaxation. As reported in Table 1, ER also stands out being one of the few methods that are fully compliant with GCL. MER [33] and GSS [1] fulfill the requirements as well, but they suffer from a very long running time which hinders their applicability to non-trivial datasets.

In this work, we propose a novel CL baseline that improves on ER while maintaining a very simple formulation. We call it **Dark Experience Replay (DER)** as it relies on *dark knowledge* [15] for distilling past *experiences*, sampled over the entire training trajectory. Our proposal satisfies the GCL guidelines and outperforms the current state-of-the-art approaches in the standard CL experiments we conduct. With respect to ER, we empirically show that our baseline exhibits remarkable qualities: it converges to flatter minima, achieves better model calibration at the cost of a limited memory and training time overhead. Eventually, we propose a novel GCL setting (MNIST-360); it displays MNIST digits sequentially and subject to a smooth increasing rotation, thus generating both sudden and gradual changes in their distribution. By evaluating the few GCL-compatible methods on MNIST-360, we show that DER also qualifies as a state-of-the-art baseline for future studies on this setting.

## 2 Related Work

*Rehearsal-based methods* tackle catastrophic forgetting by replaying a subset of the training data stored in a memory buffer. Early works [31, 34] proposed **Experience Replay (ER)**, that is interleaving old samples with current data in training batches. Several recent studies directly expand on this idea: **Meta-Experience Replay (MER)** [33] casts replay as a meta-learning problem to maximize transfer from past tasks while minimizing interference; **Gradient based Sample Selection (GSS)** [1] introduces a variation on ER to store optimally chosen examples in the memory buffer; **Hindsight Anchor Learning (HAL)** [8] complements replay with an additional objective to limit forgetting on pivotal learned data-points. On the other hand, **Gradient Episodic Memory (GEM)** [27] and its lightweight counterpart **Averaged-GEM (A-GEM)** [9] leverage old training data to build optimization constraints to be satisfied by the current update step. These works show improvements over ER when confining the learning to a small portion of the training set (*e.g.*, 1k examples per task). However, we believe that this setting rewards *sample efficiency – i.e.*, making good use of the few shown examples – which represents a potential confounding factor for assessing catastrophic forgetting. Indeed, Section 4 reveals that the above-mentioned approaches are not consistently superior to ER when lifting these restrictions, which motivates our research in this kind of methods.

*Knowledge Distillation*. Several approaches exploit Knowledge Distillation [16] to mitigate forgetting by appointing a past version of the model as a teacher. **Learning Without Forgetting (LwF)** [24] computes a smoothed version of the current responses for the new examples at the beginning of each task, minimizing their drift during training. A combination of replay and distillation can be found in **iCaRL** [32], which employs a buffer as a training set for a *nearest-mean-of-exemplars* classifier while preventing the representation from deteriorating in later tasks via a self-distillation loss term.

*Other Approaches.* Regularization-based methods extend the loss function with a term that prevents network weights from changing, as done by **Elastic Weight Consolidation (EWC)** [20], **online EWC (oEWC)** [36], **Synaptic Intelligence (SI)** [42] and **Riemmanian Walk (RW)** [7]. Architectural methods, on the other hand, devote distinguished sets of parameters to distinct tasks. Among these, **Progressive Neural Networks (PNN)** [35] instantiates new networks incrementally as novel tasks occur, resulting in a linearly growing memory requirement. To mitigate this issue, **PackNet** [28] and **Hard Attention to the Task (HAT)** [37] share the same architecture for subsequent tasks, employing a heuristic strategy to prevent intransigence by allocating additional units when needed.

# 3   Dark Experience Replay

Formally, a CL classification problem is split in $T$ tasks; during each task $t \in \{1, ..., T\}$ input samples $x$ and their corresponding ground truth labels $y$ are drawn from an i.i.d. distribution $D_t$. A function $f$, with parameters $\theta$, is optimized on one task at a time in a sequential manner. We indicate the output logits with $h_\theta(x)$ and the corresponding probability distribution over the classes with $f_\theta(x) \triangleq \mathrm{softmax}(h_\theta(x))$. The goal is to learn how to correctly classify, at any given point in training, examples from any of the observed tasks up to the current one $t \in \{1, \ldots, t_c\}$:

$$\underset{\theta}{\mathrm{argmin}} \sum_{t=1}^{t_c} \mathcal{L}_t, \quad \text{where} \quad \mathcal{L}_t \triangleq \mathbb{E}_{(x,y)\sim D_t}\big[\ell(y, f_\theta(x))\big]. \tag{1}$$

This is especially challenging as data from previous tasks are assumed to be unavailable, meaning that the best configuration of $\theta$ w.r.t. $\mathcal{L}_{1...t_c}$ must be sought without $D_t$ for $t \in \{1, \ldots, t_c - 1\}$. Ideally, we look for parameters that fit the current task well while approximating the behavior observed in the old ones: effectively, we encourage the network to mimic its original responses for past samples. To preserve the knowledge about previous tasks, we seek to minimize the following objective:

$$\mathcal{L}_{t_c} + \alpha \sum_{t=1}^{t_c-1} \mathbb{E}_{x\sim D_t}\big[D_{KL}(f_{\theta_t^*}(x) \,\|\, f_\theta(x))\big], \tag{2}$$

where $\theta_t^*$ is the optimal set of parameters at the end of task $t$, and $\alpha$ is a hyper-parameter balancing the trade-off between the terms. This objective, which resembles the teacher-student approach, would require the availability of $D_t$ for previous tasks. To overcome such a limitation, we introduce a replay buffer $\mathcal{M}_t$ holding past *experiences* for task $t$. Differently from other rehearsal-based methods [1, 8, 33], we retain the network's logits $z \triangleq h_{\theta_t}(x)$, instead of the ground truth labels $y$.

$$\mathcal{L}_{t_c} + \alpha \sum_{t=1}^{t_c-1} \mathbb{E}_{(x,z)\sim \mathcal{M}_t}\big[D_{KL}(\mathrm{softmax}(z) \,\|\, f_\theta(x))\big]. \tag{3}$$

As we focus on General Continual Learning, we intentionally avoid relying on task boundaries to populate the buffer as the training progresses. Therefore, in place of the common task-stratified sampling strategy, we adopt *reservoir* sampling [40]: this way, we select $|\mathcal{M}|$ random samples from the input stream, guaranteeing that they have the same probability $|\mathcal{M}|/|\mathcal{S}|$ of being stored in the buffer, without knowing the length of the stream $\mathcal{S}$ in advance. We can rewrite Eq. 3 as follows:

$$\mathcal{L}_{t_c} + \alpha\, \mathbb{E}_{(x,z)\sim \mathcal{M}}\big[D_{KL}(\mathrm{softmax}(z) \,\|\, f_\theta(x))\big]. \tag{4}$$

Such a strategy implies picking logits $z$ during the optimization trajectory, so potentially different from the ones that can be observed at the task's local optimum. Even if counter-intuitive, we empirically observed that this strategy does not hurt performance, while still being suitable without task boundaries. Furthermore, we observe that the replay of sub-optimal logits has beneficial effects in terms of flatness of the attained minima and calibration (see Section 5).

Under mild assumptions [16], the optimization of the KL divergence in Eq. 4 is equivalent to minimizing the Euclidean distance between the corresponding pre-softmax responses (*i.e.* logits). In this work we opt for matching logits, as it avoids the information loss occurring in probability space due to the squashing function (e.g., softmax) [26]. With these considerations in hands, Dark Experience Replay (DER, algorithm 1) optimizes the following objective:

$$\mathcal{L}_{t_c} + \alpha\, \mathbb{E}_{(x,z)\sim \mathcal{M}}\big[\|z - h_\theta(x)\|_2^2\big]. \tag{5}$$

We approximate the expectation by computing gradients on batches sampled from the replay buffer.

***Dark Experience Replay++.*** It is worth noting that the *reservoir* strategy may weaken DER under some specific circumstances. Namely, when a sudden distribution shift occurs in the input stream, logits that are highly biased by the training on previous tasks might be sampled for later replay: leveraging the ground truth labels as well – as done by ER – could mitigate such a shortcoming. On these grounds, we also propose **Dark Experience Replay++ (DER++, algorithm 2)**, which equips the objective of Eq. 5 with an additional term on buffer datapoints, promoting higher conditional likelihood w.r.t. their ground truth labels with a minimal memory overhead:

$$\mathcal{L}_{t_c} + \alpha\, \mathbb{E}_{(x',y',z')\sim \mathcal{M}}\big[\|z' - h_\theta(x')\|_2^2\big] + \beta\, \mathbb{E}_{(x'',y'',z'')\sim \mathcal{M}}\big[\ell(y'', f_\theta(x''))\big], \tag{6}$$

where $\beta$ is an additional coefficient balancing the last term[1] (DER++ collapses to DER when $\beta = 0$).

| **Algorithm 1** - Dark Experience Replay | **Algorithm 2** - Dark Experience Replay ++ |
|---|---|
| **Input:** dataset $D$, parameters $\theta$, scalar $\alpha$, learning rate $\lambda$ | **Input:** dataset $D$, parameters $\theta$, scalars $\alpha$ and $\beta$, learning rate $\lambda$ |

**Algorithm 1** - Dark Experience Replay

**Input:** dataset $D$, parameters $\theta$, scalar $\alpha$,
      learning rate $\lambda$

$\mathcal{M} \leftarrow \{\}$
**for** $(x, y)$ **in** $D$ **do**
   $(x', z', y') \leftarrow sample(\mathcal{M})$
   $x_t \leftarrow augment(x)$
   $x'_t \leftarrow augment(x')$
   $z \leftarrow h_\theta(x_t)$
   $reg \leftarrow \alpha \left\| z' - h_\theta(x'_t) \right\|_2^2$
   $\theta \leftarrow \theta + \lambda \cdot \nabla_\theta[\ell(y, f_\theta(x_t)) + reg]$
   $\mathcal{M} \leftarrow reservoir(\mathcal{M}, (x, z))$
**end for**

**Algorithm 2** - Dark Experience Replay ++

**Input:** dataset $D$, parameters $\theta$, scalars $\alpha$ and $\beta$,
      learning rate $\lambda$

$\mathcal{M} \leftarrow \{\}$
**for** $(x, y)$ **in** $D$ **do**
   $(x', z', y') \leftarrow sample(\mathcal{M})$
   $(x'', z'', y'') \leftarrow sample(\mathcal{M})$
   $x_t \leftarrow augment(x)$
   $x'_t, x''_t \leftarrow augment(x'),\ augment(x'')$
   $z \leftarrow h_\theta(x_t)$
   $reg \leftarrow \alpha \left\| z' - h_\theta(x'_t) \right\|_2^2 + \beta\, \ell(y'', f_\theta(x''_t))$
   $\theta \leftarrow \theta + \lambda \cdot \nabla_\theta[\ell(y, f_\theta(x_t)) + reg]$
   $\mathcal{M} \leftarrow reservoir(\mathcal{M}, (x, z, y))$
**end for**

## 3.1 Relation with previous works

While both our proposal and LWF [24] leverage knowledge distillation in Continual Learning, they adopt remarkably different approaches. The latter does not replay past examples, so it only encourages the similarity between teacher and student responses w.r.t. to data points of the current task. Alternatively, iCaRL [32] distills knowledge for past outputs w.r.t. past exemplars, which is more akin to our proposal. However, the former exploits the network appointed at the end of each task as the sole teaching signal. On the contrary, our methods store logits sampled throughout the optimization trajectory, which resembles having several different teacher parametrizations.

A close proposal to ours is given by **Function Distance Regularization (FDR)** for combatting catastrophic forgetting (Sec. 3.1 of [4]). Like FDR, we use past exemplars and network outputs to align past and current outputs. However, similarly to the iCaRL discussion above, FDR stores network responses at task boundaries and thus cannot be employed in a GCL setting. Instead, the experimental analysis we present in Sec. 5 reveals that the need of task boundaries can be relaxed through *reservoir* without experiencing a drop in performance; on the contrary we empirically observe that DER and DER++ achieve significantly superior results and remarkable properties. We finally highlight that the motivation behind [4] lies chiefly in studying how the training trajectory of NNs can be characterized in a functional $L^2$ Hilbert space, whereas the potential of function-space regularization for Continual Learning problems is only coarsely addressed with a single experiment on MNIST. In this respect, we present extensive experiments on multiple CL settings as well as a detailed analysis (Sec. 5) providing a deeper understanding on the effectiveness of this kind of regularization.

## 4 Experiments

We adhere to [17, 39] and model the sequence of tasks according to the following three settings:

**Task Incremental Learning (Task-IL)** and **Class Incremental Learning (Class-IL)** split the training samples into partitions of classes (tasks). Although similar, the former provides task identities to select the relevant classifier for each example, whereas the latter does not; this difference makes Task-IL and Class-IL the easiest and hardest scenarios among the three [39]. In practice, we follow [10, 42] by splitting CIFAR-10 [21] and Tiny ImageNet [38] in 5 and 10 tasks, each of which introduces 2 and 20 classes respectively. We show all the classes in the same fixed order across different runs.

**Domain Incremental Learning (Domain-IL)** feeds all classes to the network during each task, but applies a task-dependent transformation to the input; task identities remain unknown at test time. For this setting, we leverage two common protocols built upon the MNIST dataset [23], namely **Permuted MNIST** [20] and **Rotated MNIST** [27]. They both require the learner to classify all MNIST digits for 20 subsequent tasks, but the former applies a random permutation to the pixels, whereas the latter rotates the images by a random angle in the interval $[0, \pi)$.

As done in previous works [11, 32, 39, 41], we provide task boundaries to the competitors demanding them at training time (*e.g.* oEWC or LwF). This choice is meant to ensure a fair comparison between our proposal – which does not need boundaries – and a broader class of methods in literature.

## 4.1 Evaluation Protocol

*Architecture.* For tests we conducted on variants of the MNIST dataset, we follow [27, 33] by employing a fully-connected network with two hidden layers, each one comprising of 100 ReLU units. For CIFAR-10 and Tiny ImageNet, we follow [32] and rely on ResNet18 [14] (not pre-trained).

*Augmentation.* For CIFAR-10 and Tiny ImageNet, we apply random crops and horizontal flips to both stream and buffer examples. We propagate this choice to competitors for fairness. It is worth noting that combining data augmentation with our regularization objective enforces an implicit consistency loss [2, 3], which aligns predictions for the same example subjected to small data transformations.

*Hyperparameter selection.* We select hyperparameters by performing a grid-search on a validation set, the latter obtained by sampling $10\%$ of the training set. For the Domain-IL scenario, we make use of the final average accuracy as the selection criterion. Differently, we perform a combined grid-search for Class-IL and Task-IL, choosing the configuration that achieves the highest final accuracy averaged on the two settings. Please refer to the supplementary materials for a detailed characterization of the hyperparameter grids we explored along with the chosen configurations.

*Training.* To provide a fair comparison among CL methods, we train all the networks using the Stochastic Gradient Descent (SGD) optimizer. Despite being interested in an online scenario, with no additional passages on the data, we reckon it is necessary to set the number of epochs per task in relation to the dataset complexity. Indeed, if even the pure-SGD baseline fails at fitting a single task with adequate accuracy, we could not properly disentangle the effects of catastrophic forgetting from those linked to underfitting — we refer the reader to the supplementary material for an experimental discussion regarding this issue. For MNIST-based settings, one epoch per task is sufficient. Conversely, we increase the number of epochs to 50 for Sequential CIFAR-10 and 100 for Sequential Tiny ImageNet respectively, as commonly done by works that test on harder datasets [32, 41, 42]. We deliberately hold batch size and minibatch size out from the hyperparameter space, thus avoiding the flaw of a variable number of update steps for different methods.

## 4.2 Experimental Results

In this section, we compare DER and DER++ against two regularization-based methods (*oEWC*, *SI*), two methods leveraging Knowledge Distillation (*iCaRL*, *LwF*[2]), one architectural method (*PNN*) and six rehearsal-based methods (*ER*, *GEM*, *A-GEM*, *GSS*, *FDR* [4], *HAL*)[3] We further provide a lower bound, consisting of *SGD* without any countermeasure to forgetting and an upper bound given by training all tasks jointly (JOINT). Table 2 reports performance in terms of average accuracy at the end of all tasks (we refer the reader to supplementary materials for other metrics as forward and backward transfer [27]). Results are averaged across ten runs, each one involving a different initialization.

DER and DER++ achieve state-of-the-art performance in almost all settings. When compared to oEWC and SI, the gap appears unbridgeable, suggesting that regularization towards old sets of parameters does not suffice to prevent forgetting. We argue that this is due to local information modeling weights importance: as it is computed in earlier tasks, it could become untrustworthy in later ones. While being computationally more efficient, LWF performs worse than SI and oEWC on average. PNN, which achieves the strongest results among non-rehearsal methods, attains lower accuracy than replay-based ones despite its memory footprint being much higher at any buffer size.

When compared to rehearsal methods, DER and DER++ show strong performance in the majority of benchmarks, especially in the Domain-IL scenario. For these problems, a shift occurs within the input domain, but not within the classes: hence, the relations among them also likely persist. As an example, if it is true that during the first task number 2's visually look like 3's, this still holds when applying rotations or permutations, as it is done in the following tasks. We argue that leveraging soft-targets in place of hard ones (ER) carries more valuable information [16], exploited by DER and DER++ to preserve the similarity structure through the data-stream. Additionally, we observe that methods resorting to gradients (GEM, A-GEM, GSS) seem to be less effective in this setting.

The gap in performance we observe in Domain-IL is also found in the Class-IL setting, as DER is remarkably capable of learning how classes from different tasks are related to each other. This is not

| Buffer | Method | S-CIFAR-10 | | S-Tiny-ImageNet | | P-MNIST | R-MNIST |
|---|---|---|---|---|---|---|---|
| | | *Class-IL* | *Task-IL* | *Class-IL* | *Task-IL* | *Domain-IL* | *Domain-IL* |
| – | JOINT | $92.20_{\pm0.15}$ | $98.31_{\pm0.12}$ | $59.99_{\pm0.19}$ | $82.04_{\pm0.10}$ | $94.33_{\pm0.17}$ | $95.76_{\pm0.04}$ |
| | SGD | $19.62_{\pm0.05}$ | $61.02_{\pm3.33}$ | $7.92_{\pm0.26}$ | $18.31_{\pm0.68}$ | $40.70_{\pm2.33}$ | $67.66_{\pm8.53}$ |
| – | oEWC [36] | $19.49_{\pm0.12}$ | $68.29_{\pm3.92}$ | $7.58_{\pm0.10}$ | $19.20_{\pm0.31}$ | $\mathbf{75.79_{\pm2.25}}$ | $\mathbf{77.35_{\pm5.77}}$ |
| | SI [42] | $19.48_{\pm0.17}$ | $68.05_{\pm5.91}$ | $6.58_{\pm0.31}$ | $36.32_{\pm0.13}$ | $65.86_{\pm1.57}$ | $71.91_{\pm5.83}$ |
| | LwF [24] | $\mathbf{19.61_{\pm0.05}}$ | $63.29_{\pm2.35}$ | $\mathbf{8.46_{\pm0.22}}$ | $15.85_{\pm0.58}$ | - | - |
| | PNN [35] | - | $\mathbf{95.13_{\pm0.72}}$ | - | $67.84_{\pm0.29}$ | - | - |
| 200 | ER [33] | $44.79_{\pm1.86}$ | $91.19_{\pm0.94}$ | $8.49_{\pm0.16}$ | $38.17_{\pm2.00}$ | $72.37_{\pm0.87}$ | $85.01_{\pm1.90}$ |
| | GEM [27] | $25.54_{\pm0.76}$ | $90.44_{\pm0.94}$ | - | - | $66.93_{\pm1.25}$ | $80.80_{\pm1.15}$ |
| | A-GEM [9] | $20.04_{\pm0.34}$ | $83.88_{\pm1.49}$ | $8.07_{\pm0.08}$ | $22.77_{\pm0.03}$ | $66.42_{\pm4.00}$ | $81.91_{\pm0.76}$ |
| | iCaRL [32] | $49.02_{\pm3.20}$ | $88.99_{\pm2.13}$ | $7.53_{\pm0.79}$ | $28.19_{\pm1.47}$ | - | - |
| | FDR [4] | $30.91_{\pm2.74}$ | $91.01_{\pm0.68}$ | $8.70_{\pm0.19}$ | $40.36_{\pm0.68}$ | $74.77_{\pm0.83}$ | $85.22_{\pm3.35}$ |
| | GSS [1] | $39.07_{\pm5.59}$ | $88.80_{\pm2.89}$ | - | - | $63.72_{\pm0.70}$ | $79.50_{\pm0.41}$ |
| | HAL [8] | $32.36_{\pm2.70}$ | $82.51_{\pm3.20}$ | - | - | $74.15_{\pm1.65}$ | $84.02_{\pm0.98}$ |
| | **DER (ours)** | $61.93_{\pm1.79}$ | $91.40_{\pm0.92}$ | $\mathbf{11.87_{\pm0.78}}$ | $40.22_{\pm0.67}$ | $81.74_{\pm1.07}$ | $90.04_{\pm2.61}$ |
| | **DER++ (ours)** | $\mathbf{64.88_{\pm1.17}}$ | $\mathbf{91.92_{\pm0.60}}$ | $10.96_{\pm1.17}$ | $\mathbf{40.87_{\pm1.16}}$ | $\mathbf{83.58_{\pm0.59}}$ | $\mathbf{90.43_{\pm1.87}}$ |
| 500 | ER [33] | $57.74_{\pm0.27}$ | $93.61_{\pm0.27}$ | $9.99_{\pm0.29}$ | $48.64_{\pm0.46}$ | $80.60_{\pm0.86}$ | $88.91_{\pm1.44}$ |
| | GEM [27] | $26.20_{\pm1.26}$ | $92.16_{\pm0.69}$ | - | - | $76.88_{\pm0.52}$ | $81.15_{\pm1.98}$ |
| | A-GEM [9] | $22.67_{\pm0.57}$ | $89.48_{\pm1.45}$ | $8.06_{\pm0.04}$ | $25.33_{\pm0.49}$ | $67.56_{\pm1.28}$ | $80.31_{\pm6.29}$ |
| | iCaRL [32] | $47.55_{\pm3.95}$ | $88.22_{\pm2.62}$ | $9.38_{\pm1.53}$ | $31.55_{\pm3.27}$ | - | - |
| | FDR [4] | $28.71_{\pm3.23}$ | $93.29_{\pm0.59}$ | $10.54_{\pm0.21}$ | $49.88_{\pm0.71}$ | $83.18_{\pm0.53}$ | $89.67_{\pm1.63}$ |
| | GSS [1] | $49.73_{\pm4.78}$ | $91.02_{\pm1.57}$ | - | - | $76.00_{\pm0.87}$ | $81.58_{\pm0.58}$ |
| | HAL [8] | $41.79_{\pm4.46}$ | $84.54_{\pm2.36}$ | - | - | $80.13_{\pm0.49}$ | $85.00_{\pm0.96}$ |
| | **DER (ours)** | $70.51_{\pm1.67}$ | $93.40_{\pm0.39}$ | $17.75_{\pm1.14}$ | $51.78_{\pm0.88}$ | $87.29_{\pm0.46}$ | $92.24_{\pm1.12}$ |
| | **DER++ (ours)** | $\mathbf{72.70_{\pm1.36}}$ | $\mathbf{93.88_{\pm0.50}}$ | $\mathbf{19.38_{\pm1.41}}$ | $\mathbf{51.91_{\pm0.68}}$ | $\mathbf{88.21_{\pm0.39}}$ | $\mathbf{92.77_{\pm1.05}}$ |
| 5120 | ER [33] | $82.47_{\pm0.52}$ | $\mathbf{96.98_{\pm0.17}}$ | $27.40_{\pm0.31}$ | $67.29_{\pm0.23}$ | $89.90_{\pm0.13}$ | $93.45_{\pm0.56}$ |
| | GEM [27] | $25.26_{\pm3.46}$ | $95.55_{\pm0.02}$ | - | - | $87.42_{\pm0.95}$ | $88.57_{\pm0.40}$ |
| | A-GEM [9] | $21.99_{\pm2.29}$ | $90.10_{\pm2.09}$ | $7.96_{\pm0.13}$ | $26.22_{\pm0.65}$ | $73.32_{\pm1.12}$ | $80.18_{\pm5.52}$ |
| | iCaRL [32] | $55.07_{\pm1.55}$ | $92.23_{\pm0.84}$ | $14.08_{\pm1.92}$ | $40.83_{\pm3.11}$ | - | - |
| | FDR [4] | $19.70_{\pm0.07}$ | $94.32_{\pm0.97}$ | $28.97_{\pm0.41}$ | $68.01_{\pm0.42}$ | $90.87_{\pm0.16}$ | $94.19_{\pm0.44}$ |
| | GSS [1] | $67.27_{\pm4.27}$ | $94.19_{\pm1.15}$ | - | - | $82.22_{\pm1.14}$ | $85.24_{\pm0.59}$ |
| | HAL [8] | $59.12_{\pm4.41}$ | $88.51_{\pm3.32}$ | - | - | $89.20_{\pm0.14}$ | $91.17_{\pm0.31}$ |
| | **DER (ours)** | $83.81_{\pm0.33}$ | $95.43_{\pm0.33}$ | $36.73_{\pm0.64}$ | $69.50_{\pm0.26}$ | $91.66_{\pm0.11}$ | $94.14_{\pm0.31}$ |
| | **DER++ (ours)** | $\mathbf{85.24_{\pm0.49}}$ | $96.12_{\pm0.21}$ | $\mathbf{39.02_{\pm0.97}}$ | $\mathbf{69.84_{\pm0.63}}$ | $\mathbf{92.26_{\pm0.17}}$ | $\mathbf{94.65_{\pm0.33}}$ |

Table 2: Classification results for standard CL benchmarks, averaged across 10 runs. '-' indicates experiments we were unable to run, because of compatibility issues (*e.g.* between PNN, iCaRL and LwF in Domain-IL) or intractable training time (*e.g.* GEM, HAL or GSS on Tiny ImageNet).

so relevant in Task-IL, where DER performs on par with ER on average. In it, classes only need to be compared in exclusive subsets, and maintaining an overall vision is not especially rewarding. In such a scenario, DER++ manages to effectively combine the strengths of both methods, resulting in generally better accuracy. Interestingly, iCaRL appears valid when using a small buffer; we believe that this is due to its helpful *herding* strategy, ensuring that all classes are equally represented in memory. As a side note, other ER-based methods (HAL and GSS) show weaker results than ER itself on such challenging datasets.

## 4.3 MNIST-360

To address the General Continual Learning desiderata, we propose a novel protocol: MNIST-360. It models a stream of data presenting batches of two consecutive MNIST digits at a time (*e.g.* $\{0, 1\}$, $\{1, 2\}$, $\{2, 3\}$ etc.), as depicted in Fig. 1. We rotate each example of the stream by an increasing angle and, after a fixed number of steps, switch the lesser of the two digits with the following one. As it is impossible to distinguish 6's and 9's upon rotation, we do not use 9's in MNIST-360. The stream visits the nine possible couples of classes three times, allowing the model to leverage positive transfer when revisiting a previous task. In the implementation, we guarantee that: i) each example is shown once during the overall training; ii) two digits of the same class are never observed under the same rotation. We provide a detailed description of training and test sets in supplementary materials.

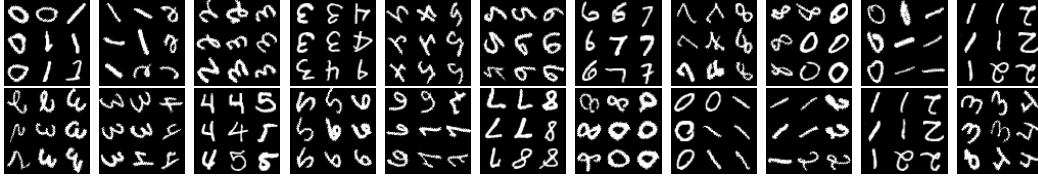

Figure 1: Example batches of the MNIST-360 stream.

| JOINT | SGD | Buffer | ER [33] | MER [33] | A-GEM-R [9] | GSS [1] | DER (ours) | DER++ (ours) |
|---|---|---|---|---|---|---|---|---|
| $82.98_{\pm 3.24}$ | $19.09_{\pm 0.69}$ | 200 | $49.27_{\pm 2.25}$ | $48.58_{\pm 1.07}$ | $28.34_{\pm 2.24}$ | $43.92_{\pm 2.43}$ | $\mathbf{55.22_{\pm 1.67}}$ | $54.16_{\pm 3.02}$ |
| | | 500 | $65.04_{\pm 1.53}$ | $62.21_{\pm 1.36}$ | $28.13_{\pm 2.62}$ | $54.45_{\pm 3.14}$ | $69.11_{\pm 1.66}$ | $\mathbf{69.62_{\pm 1.59}}$ |
| | | 1000 | $75.18_{\pm 1.50}$ | $70.91_{\pm 0.76}$ | $29.21_{\pm 2.62}$ | $63.84_{\pm 2.09}$ | $75.97_{\pm 2.08}$ | $\mathbf{76.03_{\pm 1.61}}$ |

Table 3: Accuracy on the test set for MNIST-360.

It is worth noting that such a setting presents both sharp (change in class) and smooth (rotation) distribution shifts; therefore, for the algorithms that rely on explicit boundaries, it would be hard to identify them. As outlined in Section 1, just ER, MER, and GSS are suitable for GCL. However, we also explore a variant of A-GEM equipped with a reservoir memory buffer (A-GEM-R). We compare these approaches with DER and DER++, reporting the results in Table 3 (we keep the same fully-connected network we used on MNIST-based datasets). As can be seen, DER and DER++ outperform other approaches in such a challenging scenario, supporting the effectiveness of the proposed baselines against alternative replay methods. Due to space constraints, we refer the reader to supplementary materials for an additional evaluation regarding the memory footprint.

## 5 Model Analysis

In this section, we provide an in depth analysis of DER and DER++ by comparing them against FDR and ER. By so doing, we gather insights on the employment of logits sampled throughout the optimization trajectory, as opposed to ones at task boundaries and ground truth labels.

***DER converges to flatter minima.*** Recent studies [6, 18, 19] link Deep Network generalization to the geometry of the loss function, namely the flatness of the attained minimum. While these works link flat minima to good train-test generalization, here we are interested in examining their weight in Continual Learning. Let us suppose that the optimization converges to a sharp minimum w.r.t. $\mathcal{L}_{1...t_c}$ (Eq. 1): in that case, the tolerance towards local perturbations is quite low. As a side effect, the drift we will observe in parameter space (due to the optimization of $\mathcal{L}_{1...t'}$ for $t' > t_c$) will intuitively lead to an even more serious drop in performance.

On the contrary, reaching a flat minimum for $\mathcal{L}_{1...t_c}$ could give more room for exploring neighbouring regions of the parameter space, where one may find a new optimum for task $t'$ without experiencing a severe failure on tasks $1, \ldots, t_c$. We conjecture that the effectiveness of the proposed baseline is linked to its ability to attain flatter and robust minima, which generalizes better to unseen data and, additionally, favors adaptability to incoming tasks. To validate this hypothesis, we compare the flatness of the training minima of FDR, ER, DER and DER++ utilizing two distinct metrics.

Firstly, as done in [43, 44], we consider the model at the end of training and add independent Gaussian noise with growing $\sigma$ to each parameter. This allows us to evaluate its effect on the average loss across all training examples. As shown in Fig. 2(a) (S-CIFAR-10, buffer size 500), ER and especially FDR reveal higher sensitivity to perturbations than DER and DER++. Furthermore, [6, 18, 19] propose measuring flatness by evaluating the eigenvalues of $\nabla_\theta^2 \mathcal{L}$: sharper minima correspond to larger Hessian eigenvalues. At the end of training on S-CIFAR-10, we compute the empirical Fisher Information Matrix $F = \sum \nabla_\theta \mathcal{L} \nabla_\theta \mathcal{L}^T / N$ w.r.t. the whole training set (as an approximation of the intractable Hessian [6, 20]). Fig. 2(b) reports the sum of its eigenvalues $\mathrm{Tr}(F)$: as one can see, DER and especially DER++ produce the lowest eigenvalues, which translates into flatter minima following our intuitions. It is worth noting that FDR's large $\mathrm{Tr}(F)$ for buffer size $5120$ could be linked to its failure case in S-CIFAR-10, Class-IL.

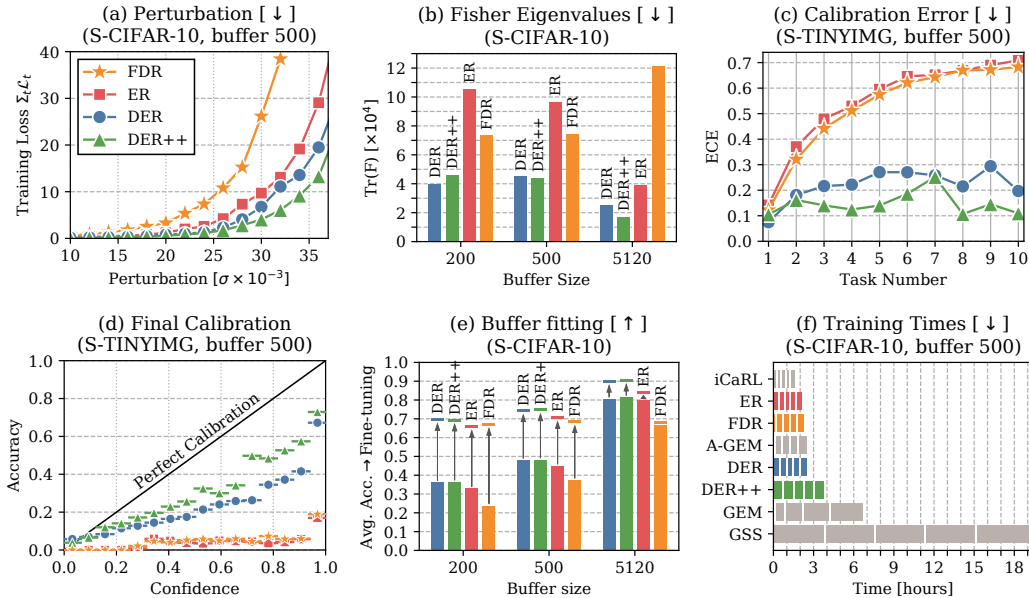

Figure 2: Results for the model analysis. [↑] higher is better, [↓] lower is better *(best seen in color)*.

***DER converges to more calibrated networks.*** Calibration is a desirable property for a learner, measuring how much the confidence of its predictions corresponds to its accuracy. Ideally, we expect output distributions whose shapes mirror the probability of being correct, thus quantifying how much one can trust a specific prediction. Recent works find out that modern Deep Networks – despite largely outperforming the ones from a decade ago – are less calibrated [13], as they tend to yield overconfident predictions [22]. In real-world applications, AI tools should support decisions in a continuous and online fashion (*e.g.* weather forecasting [5] or econometric analysis [12]); therefore, calibration represents an appealing property for any CL system aiming for employment outside of a laboratory environment.

Fig. 2(c, d) shows, for TinyImageNet, the value of the Expected Calibration Error (ECE) [30] during the training and the reliability diagram at the end of it respectively. It can be seen that DER and DER++ achieve a lower ECE than ER and FDR without further application of *a posteriori* calibration methods (*e.g.*, Temperature Scaling, Dirichlet Calibration, ...). This means that models trained using Dark Experience are less overconfident and, therefore, easier to interpret. As a final remark, *Liu et al.* link this property to the capability to generalize to novel classes in a zero-shot scenario [25], which could translate into an advantageous starting point for the subsequent tasks for DER and DER++.

***On the informativeness of DER's buffer.*** Network responses provide a rich description of the corresponding data point. Following this intuition, we posit that the merits of DER also result from the knowledge inherent in its memory buffer: when compared to the one built by ER, the former represents a more informative summary of the overall (full) CL problem. If that were the case, a new learner trained only on the buffer would yield an accuracy that is closer to the one given by jointly training on all data. To validate this idea, we train a network from scratch using the memory buffer as the training set: we can hence compare how memories produced by DER, ER, and FDR summarize well the underlying distribution. Fig. 2(e) shows the accuracy on the test set: as can be seen, DER delivers the highest performance, surpassing ER, and FDR. This is particularly evident for smaller buffer sizes, indicating that DER's buffer should be especially preferred in scenarios with severe memory constraints.

Further than its pure performance, we assess whether a model trained on the buffer can be specialized to an already seen task: this would be the case of new examples from an old distribution becoming available on the stream. We simulate it by sampling 10 samples per class from the test set and then fine-tuning on them with no regularization; Fig. 2 reports the average accuracy on the remainder of the test set of each task: here too, DER's buffer yields better performance than ER and FDR, thus providing additional insight regarding its representation capabilities.

***On training time.*** When facing up with a data-stream, one often cares about reducing the overall processing time: otherwise, training would not keep up with the rate at which data are made available to the stream. In this regard, we assess the performance of both DER and DER++ and other rehearsal methods in terms of wall-clock time (seconds) at the end of the last task. To guarantee a fair comparison, we conduct all tests under the same conditions, running each benchmark on a Desktop Computer equipped with an NVIDIA Titan X GPU and an Intel i7-6850K CPU. Fig. 2(f) reports the execution time we measured on S-CIFAR10, indicating the time necessary for each of 5 tasks. We draw the following remarks: i) DER has a comparable running time w.r.t. other replay methods such as ER, FDR, and A-GEM; ii) the time complexity for GEM grows linearly w.r.t. the number of previously seen tasks; iii) GSS is extremely slow (0.73 examples per second on average, while DER++ processes 3.71 examples per second), making it hardly viable in practical scenarios.

## 6   Conclusions

In this paper, we introduce Dark Experience Replay: a simple baseline for Continual Learning, which leverages Knowledge Distillation for retaining past experience and therefore avoiding catastrophic forgetting. We show the effectiveness of our proposal through an extensive experimental analysis, carried out on top of standard benchmarks. Also, we argue that the recently formalized General Continual Learning provides the foundation for advances in diverse applications; for this reason, we propose MNIST-360 as an experimental protocol for this setting. We recommend DER as a starting point for future studies on both CL and GCL in light of its strong results on all evaluated settings and of the properties observed in Sec. 5.

## Broader Impact

We hope that this work will prove useful to the Continual Learning (CL) scientific community as it is fully reproducible and includes:

- a clear and extensive comparison of the state of the art on multiple datasets;
- Dark Experience Replay (DER), a simple baseline that outperforms all other methods while maintaining a limited memory footprint.

As revealed by the analysis in Section 5, DER also proves to be better calibrated than a simple Experience Replay baseline, which means that it could represent a useful starting point for the study of CL decision-making applications where an overconfident model would be detrimental.

We especially hope that the community will benefit from the introduction of MNIST-360, the first evaluation protocol adhering to the General Continual Learning scenario. The latter has been recently proposed to describe the requirement of a CL system that can be applied to real-world problems. Widespread adoption of our protocol (or new ones of similar design) can close the gap between the current CL studies and practical AI systems. Due to the abstract nature of MNIST-360 (it only contains digits), we believe that ethical and bias concerns are not applicable.

## Acknowledgments and Disclosure of Funding

This research was funded by the Artificial Intelligence Research and Innovation Center (AIRI) within the University of Modena and Reggio Emilia. We also acknowledge Panasonic Silicon Valley Lab, NVIDIA Corporation and Facebook Artificial Intelligence Research for their contribution to the hardware infrastructure we used for our experiments.

## Footnotes

[1]The model is not overly sensitive to $\alpha$ and $\beta$: setting them both to $0.5$ yields stable performance.

[2]In Class-IL, we adopted a multi-class implementation as done in [32].

[3]We omit MER as we experienced an intractable training time on these benchmarks (e.g. while DER takes approximately 2.5 hours on Seq. CIFAR-10, MER takes 300 hours – see Sec. 5 for further comparisons).

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
