[Supplementary Material · supplementary.pdf]

# Dark Experience for General Continual Learning: a Strong, Simple Baseline
# - Supplementary Material -

**Pietro Buzzega**    **Matteo Boschini**    **Angelo Porrello**    **Davide Abati**    **Simone Calderara**

AImageLab - University of Modena and Reggio Emilia, Modena, Italy
`name.surname@unimore.it`

## A   Justification of Table 1

Below we provide a justification for each mark of Table 1:

### A.1   Constant Memory

- *Distillation methods* need to accommodate a teacher model along with the current learner, at a fixed memory cost. While iCaRL maintains a snapshot of the network as teacher, LWF stores teacher responses to new task data at the beginning of each task.
- *Rehearsal methods* need to store a memory buffer of a fixed size. This also affects iCaRL.
- *Architectural methods* increase the model size linearly with respect to the number of tasks. In more detail, PackNet and HAT need a Boolean and float mask respectively, while PNN devotes a whole new network to each task.
- *Regularization methods* usually require to store up to two parameters sets, thus respecting the constant memory constraint.

### A.2   No Task Boundaries

- *Distillation methods* depend on task boundaries to appoint a new teacher. iCaRL also depends on them to update its memory buffer, in accordance with the *herding* sampling strategy.
- *Architectural methods* require to know exactly when the task finishes to update the model. PackNet also re-trains the network at this time.
- *Regularization methods* exploit the task change to take a snapshot of the network, using it to constrain drastic changes for the most important weights (oEWC, SI). Online EWC also needs to pass over the whole training set to compute the weights importance.
- *Rehearsal methods* can operate in the absence of task boundaries if their memory buffer exploits to the *reservoir* sampling strategy. This applies to ER, GSS, MER, DER and DER++ and can easily be extended to A-GEM (by replacing *ring* sampling with *reservoir* as discussed in Sec. 4.3). Other rehearsal approaches, however, rely on boundaries to perform specific steps: HAL hallucinates new anchors that synthesize the task it just completed, whereas FDR needs them to record converged logits to replay.

GEM does not strictly depend on task boundaries, but rather on task identities to associate every memorized input with its original task (as described in Sec. 3 of [7]). This is meant to let GEM set up a separate QP for each past task (notice that this is instead unnecessary for A-GEM, which only solves one generic constraint w.r.t. the average gradient of all buffer items). We acknowledge

that reliance on task boundaries and reliance on task identities are *logically equivalent*: indeed, i) the availability of task identities straightforwardly allows any method to recognize and exploit task boundaries; ii) *vice versa*, by relying on task boundaries and maintaining a task counter, one can easily associate task identities to incoming input points (under the assumption that tasks are always shown in a sequence without repetitions). This explains why Table 1 indicates that GEM depends on task boundaries. This is also in line with what argued by the authors of [1].

### A.3 No test time oracle

- *Architectural methods* need to know the task label to modify the model accordingly before they make any prediction.
- LWF is designed as a multi-head method, which means that its prediction head must be chosen in accordance with the task label.
- *Regularization methods*, *rehearsal methods* and iCaRL can perform inference with no information about the task.

## B Details on the Implementation of MNIST-360

MNIST-360 presents the evaluated method with a sequence of MNIST digits from $0$ to $8$ shown at increasing angles.

### B.1 Training

For Training purposes, we build batches using exemplars that belong to two consequent classes at a time, meaning that 9 pairs of classes are possibly encountered: $(0,1)$, $(1,2)$, $(2,3)$, $(3,4)$, $(4,5)$, $(5,6)$, $(6,7)$, $(7,8)$, and $(8,0)$. Each pair is shown in this order in $R$ rounds ($R = 3$ in our experiments) at changing rotations. This means that MNIST-360 consists of $9 \cdot R$ pseudo-tasks, whose boundaries are not signaled to the tested method. We indicate them with $\Psi_r^{(d_1,d_2)}$ where $r \in \{1, \ldots, R\}$ is the round number and $d_1, d_2$ are digits forming one of the pairs listed above.

As every MNIST digit $d$ appears in $2 \cdot R$ pseudo-tasks, we randomly split its example images evenly in 6 groups $G_i^d$ where $i \in \{1, \ldots, 2 \cdot R\}$. The set of exemplars that are shown in $\Psi_r^{(d_1,d_2)}$ is given as $G_{[r/2]}^{d_1} \cup G_{[r/2]+1}^{d_2}$, where $[r/2]$ is an integer division.

At the beginning of $\Psi_r^{(d_1,d_2)}$, we initialize two counters $C_{d_1}$ and $C_{d_2}$ to keep track of how many exemplars of $d_1$ and $d_2$ are shown respectively. Given batch size $B$ ($B = 16$ in our experiments), each batch is made up of $N_{d_1}$ samples from $G_{[r/2]}^{d_1}$ and $N_{d_2}$ samples from $G_{[r/2]+1}^{d_2}$, where:

$$N_{d_1} = min \left( \frac{|G_{[r/2]}^{d_1}| - C_{d_1}}{|G_{[r/2]}^{d_1}| - C_{d_1} + |G_{[r/2]+1}^{d_2}| - C_{d_2}} \cdot B, |G_{[r/2]}^{d_1}| - C_{d_1} \right) \qquad (7)$$

$$N_{d_2} = min \left( B - N_{d_1}, |G_{[r/2]+1}^{d_2}| - C_{d_2} \right) \qquad (8)$$

This allows us to produce balanced batches, in which the proportion of exemplars of $d_1$ and $d_2$ is maintained the same. Pseudo-task $\Psi_r^{(d_1,d_2)}$ ends when the entirety of $G_{[r/2]}^{d_1} \cup G_{[r/2]+1}^{d_2}$ has been shown, which does not necessarily happen after a fixed number of batches.

Each digit $d$ is also associated with a counter $C_d^r$ that is never reset during training and is increased every time an exemplar of $d$ is shown to the evaluated method. Before its showing, every exemplar is rotated by

$$\frac{2\pi}{|d|} C_d^r + O_d \qquad (9)$$

where $|d|$ is the number of total examples of digit $d$ in the training set and $O_d$ is a digit-specific angular offset, whose value for the $ith$ digit is given by $O_i = (i - 1)\frac{\pi}{2 \cdot R}$ ($O_0 = -\frac{\pi}{2 \cdot R}$, $O_1 = 0$, $O_2 = \frac{\pi}{2 \cdot R}$, etc.). By so doing, every digit's exemplars are shown with an increasing rotation spanning an entire $2\pi$ angle throughout the entire procedure. Rotation changes within each pseudo-task, resulting into a gradually changing distribution. Fig. 1 in the main paper shows the first batch of the initial 11 pseudo-tasks with $B = 9$.

## B.2 Test

As no task boundaries are provided, evaluation on MNIST-360 can only be carried out after the training is complete. For test purposes, digits are still shown with an increasing rotation as per Eq. 9, with $|d|$ referring to the test-set digit cardinality and no offset applied ($O_d = 0$).

The order with which digits are shown is irrelevant, therefore no specific batching strategy is necessary and we simply show one digit at a time.

## C   Accuracy vs. Memory Occupation

In Fig. 3, we show how the accuracy results for the experiments in Section 4.2 and F.1 relate to the total memory usage of the evaluated methods. We maintain that having a reduced memory footprint is especially important for a CL method. This is usually fairly easy to assess for rehearsal-based methods, as they clearly specify the number of items that must be saved in the memory buffer. While this could lead to the belief that they have higher memory requirements than other classes of solutions [4], it should be noted that architectural, distillation- and regularization-based methods can instead be characterized by non-negligible fixed overheads, making them less efficient and harder to scale.

Figure 3: Performance vs. memory allocation for the experiments of Section 4 and F.1. Successive points of the same method indicate increasing buffer size. Methods with lower accuracy or excessive memory consumption may be omitted *(best viewed in color).*

## D  Reservoir Sampling Algorithm

In the following, we provide the buffer insertion algorithm (3) for the Reservoir Sampling strategy [9].

---

**Algorithm 3** - Reservoir Sampling

**Input:** memory buffer $\mathcal{M}$, number of seen examples $N$, example $x$, label $y$.
**if** $\mathcal{M} > N$ **then**
    $\mathcal{M}[N] \leftarrow (x, y)$
**else**
    $j = randomInteger \, (\min = 0, \max = N)$
    **if** $j < |\mathcal{M}|$ **then**
        $\mathcal{M}[j] \leftarrow (x, y)$
    **end if**
**end if**
**return** $M$

---

## E  Details on the Implementation of iCaRL

Although iCaRL [8] was initially proposed for the Class-IL setting, we make it possible to use it for Task-IL as well by introducing a modification of its classification rule. Let $\mu_y$ be the average feature vector of the exemplars for class $y$ and $\phi(x)$ be the feature vector computed on example $x$, iCaRL predicts a label $y^*$ as

$$y^* = \underset{y=1,\dots,t}{\arg\min} \, \|\phi(x) - \mu_y\|. \tag{10}$$

Instead, given the tensor of average feature vectors for all classes $\boldsymbol{\mu}$, we formulate iCaRL's network response $h(x)$ as

$$h(x) = -\|\phi(x) - \boldsymbol{\mu}\|. \tag{11}$$

Considering the argmax for $h(x)$, without masking (Class-IL setting), results in the same prediction as Eq. 10.

It is also worth noting that iCaRL exploits a weight-decay regularization term (*wd_reg*), as suggested in [8], in order to make its performance competitive with the other proposed approaches.

# F  Additional Results

## F.1  Sequential-MNIST

Similarly to Sequential CIFAR-10, the Sequential MNIST protocol split the whole training set of the MNIST Digits dataset in 5 tasks, each of which introduces two new digits.

| Model | Class-IL | | | Task-IL | | |
|---|---|---|---|---|---|---|
| JOINT | $95.57_{\pm0.24}$ | | | $99.51_{\pm0.07}$ | | |
| SGD | $19.60_{\pm0.04}$ | | | $94.94_{\pm2.18}$ | | |
| oEWC | $\mathbf{20.46_{\pm1.01}}$ | | | $98.39_{\pm0.48}$ | | |
| SI | $19.27_{\pm0.30}$ | | | $96.00_{\pm2.04}$ | | |
| LwF | $19.62_{\pm0.01}$ | | | $94.11_{\pm3.01}$ | | |
| PNN | - | | | $\mathbf{99.23_{\pm0.20}}$ | | |
| Buffer | 200 | 500 | 5120 | 200 | 500 | 5120 |
| ER | $80.43_{\pm1.89}$ | $86.12_{\pm1.89}$ | $93.40_{\pm1.29}$ | $97.86_{\pm0.35}$ | $\mathbf{99.04_{\pm0.18}}$ | $99.33_{\pm0.22}$ |
| MER | $81.47_{\pm1.56}$ | $88.35_{\pm0.41}$ | $94.57_{\pm0.18}$ | $98.05_{\pm0.25}$ | $98.43_{\pm0.11}$ | $99.27_{\pm0.09}$ |
| GEM | $80.11_{\pm1.54}$ | $85.99_{\pm1.35}$ | $95.11_{\pm0.87}$ | $97.78_{\pm0.25}$ | $98.71_{\pm0.20}$ | $99.44_{\pm0.12}$ |
| A-GEM | $45.72_{\pm4.26}$ | $46.66_{\pm5.85}$ | $54.24_{\pm6.49}$ | $98.61_{\pm0.24}$ | $98.93_{\pm0.21}$ | $98.93_{\pm0.20}$ |
| iCaRL | $70.51_{\pm0.53}$ | $70.10_{\pm1.08}$ | $70.60_{\pm1.03}$ | $98.28_{\pm0.09}$ | $98.32_{\pm0.07}$ | $98.32_{\pm0.11}$ |
| FDR | $79.43_{\pm3.26}$ | $85.87_{\pm4.04}$ | $87.47_{\pm3.15}$ | $97.66_{\pm0.18}$ | $97.54_{\pm1.90}$ | $97.79_{\pm1.33}$ |
| GSS | $38.90_{\pm2.49}$ | $49.76_{\pm4.73}$ | $89.39_{\pm0.75}$ | $95.02_{\pm1.85}$ | $97.71_{\pm0.53}$ | $98.33_{\pm0.17}$ |
| HAL | $84.70_{\pm0.87}$ | $87.21_{\pm0.49}$ | $89.52_{\pm0.96}$ | $97.96_{\pm0.21}$ | $98.03_{\pm0.22}$ | $98.35_{\pm0.17}$ |
| **DER (ours)** | $84.55_{\pm1.64}$ | $90.54_{\pm1.18}$ | $94.90_{\pm0.57}$ | $\mathbf{98.80_{\pm0.15}}$ | $98.84_{\pm0.13}$ | $99.29_{\pm0.11}$ |
| **DER++ (ours)** | $\mathbf{85.61_{\pm1.40}}$ | $\mathbf{91.00_{\pm1.49}}$ | $\mathbf{95.30_{\pm1.20}}$ | $98.76_{\pm0.28}$ | $98.94_{\pm0.27}$ | $\mathbf{99.47_{\pm0.07}}$ |

Table 4: Results for the Sequential-MNIST dataset.

## F.2  Additional Comparisons with Experience Replay

Figure 4: A comparison between our proposal (DER++) and the variants of Experience Replay presented in [5].

In the main paper we already draw a thorough comparison with Experience Replay (ER), showing that DER and DER++ often result in better performance and more remarkable capabilities. It is worth

noting that the ER we compared with was equipped with the *reservoir* strategy; therefore, it would be interesting to see whether the same experimental conclusions also hold for other variants of naive replay (*e.g.* ER with ring-buffer). For this reason, Fig. 4 provides further analysis in the setting of [5], which investigates what happens when varying the number of samples that are retained for later replay. Interestingly, while *reservoir* weakens ER when very few past memories are available, it does not bring DER++ to the same flaw. In the low-memory regime, indeed, the probability of leaving a class out from the buffer increases: while ER would not have any chance to retain the knowledge underlying these "ghost" classes, we conjecture that DER++ could recover this information from the non-argmax outputs of the past predicted distribution.

## F.3 Single-Epoch Setting

|  | Buffer | ER | FDR | DER++ | JOINT | JOINT |
|---|---|---|---|---|---|---|
| **#epochs** |  | 1 | 1 | 1 | 1 | 50/100 |
| Seq. CIFAR-10 | 200 | 37.64 | 21.22 | **41.93** | 56.74 | 92.20 |
|  | 500 | 45.22 | 21.06 | **48.04** |  |  |
|  | 5120 | 50.28 | 20.57 | **53.31** |  |  |
| Seq. Tiny ImageNet | 200 | 5.98 | 4.87 | **6.35** | 19.37 | 59.99 |
|  | 500 | 8.39 | 4.76 | **8.65** |  |  |
|  | 5120 | 16.04 | 4.96 | **16.41** |  |  |

Table 5: Single-epoch evaluation setting (Class-IL).

Several Continual Learning works present experiments even on fairly complex datasets (*e.g.:* CIFAR-10, CIFAR-100, Mini ImageNet) in which the model is only trained for one epoch for each task [1, 3, 4, 7]. As showing the model each example only once could be deemed closer to real-world CL scenarios, this is a very compelling setting and somewhat close in spirit to the reasons why we focus on General Continual Learning.

However, we see that committing to just one epoch (hence, few gradient steps) makes it difficult to disentangle the effects of catastrophic forgetting (the focus of our work) from those of underfitting. This is especially relevant when dealing with complex datasets and deserves further investigation: for this reason, we conduct a single-epoch experiment on Seq. CIFAR-10 and Seq. Tiny ImageNet. We include in Tab. 5 the performance of different rehearsal methods; additionally, we report the results of joint training when limiting the number of epochs to one and, *vice versa*, when such limitation is removed (see last two columns). While the multi-epoch joint training learns to classify with a satisfactory accuracy, the single-epoch counterpart (which is the upper bound to all other methods in this experiment) yields a much lower accuracy and underfits dramatically. In light of this, it is hard to evaluate the merits of other CL methods, whose evaluation is severely undermined by this confounding factor. Although DER++ proves reliable even in this difficult setting, we feel that future CL works should strive for realism by designing experimental settings which are fully in line with the guidelines of GCL [6] rather than adopting the single-epoch protocol.

## F.4 Forward and Backward Transfer

In this section, we present additional results for the experiments presented in Sec. 4.2 and F.1, reporting *Forward Transfer* (FWT), *Backward Transfer* (BWT) [7] and *Forgetting* (FRG) [2]. The first one assesses whether a model is capable of improving on unseen tasks w.r.t. random guessing, whereas the second and third ones measure the performance degradation in subsequent tasks. Despite their popularity in recent CL works [2, 3, 6, 7], we did not report them in the main paper because we believe that the average accuracy represents an already exhaustive measure of CL performance.

FWT is computed as the difference between the accuracy just before starting training on a given task and the one of the random-initialized network; it is averaged across all tasks. While one can argue that learning to classify unseen classes is desirable, the meaning of such a measure is highly dependent on the setting. Indeed, Class-IL and Task-IL show distinct classes in distinct tasks, which makes transfer impossible. On the contrary, FWT can be relevant for Domain-IL scenarios, provided

that the input transformation is not disruptive (as it is the case with Permuted-MNIST). In conclusion, as CL settings sooner or later show all classes to the network, we are primarily interested in the accuracy at the end of the training, not the one before seeing any example.

FRG and BWT compute the difference between the current accuracy and its best value for each task. It is worth noting that any method that restrains the learning of the current task could exhibit high backward transfer but low final accuracy. This is as easy as increasing the weight of the regularization term: this way, the past knowledge is well-preserved but the current task is not learned properly. Moreover, BWT makes the assumption that the highest value of the accuracy on a task is the one yielded at the end of it. This is not always true, as rehearsal-based methods can exploit the memory buffer in a subsequent task, even enhancing their performance on a previous one if they start from low accuracy.

| | | FORWARD TRANSFER | | | | | |
|---|---|---|---|---|---|---|---|
| | | **S-MNIST** | | **S-CIFAR-10** | | **P-MNIST** | **R-MNIST** |
| **Buffer** | **Method** | *Class-IL* | *Task-IL* | *Class-IL* | *Task-IL* | *Domain-IL* | *Domain-IL* |
| – | SGD | $-11.06_{\pm2.90}$ | $2.33_{\pm4.71}$ | $-9.09_{\pm0.11}$ | $-1.46_{\pm1.17}$ | $0.32_{\pm0.85}$ | $48.94_{\pm0.10}$ |
| – | oEWC | $\mathbf{-7.44_{\pm4.18}}$ | $\mathbf{-0.13_{\pm8.12}}$ | $-12.51_{\pm0.02}$ | $-4.09_{\pm7.97}$ | $0.69_{\pm0.97}$ | $52.45_{\pm8.75}$ |
| | SI | $-9.50_{\pm5.27}$ | $-1.34_{\pm5.42}$ | $-12.64_{\pm0.20}$ | $-2.33_{\pm2.29}$ | $\mathbf{0.71_{\pm1.89}}$ | $\mathbf{53.09_{\pm0.73}}$ |
| | LwF | $-12.39_{\pm4.06}$ | $1.30_{\pm5.40}$ | $\mathbf{-10.63_{\pm5.12}}$ | $0.73_{\pm4.36}$ | - | - |
| | PNN | - | N/A | - | N/A | - | - |
| 200 | ER | $-12.12_{\pm2.21}$ | $-0.86_{\pm3.24}$ | $-11.02_{\pm2.77}$ | $2.10_{\pm1.27}$ | $1.37_{\pm0.48}$ | $66.79_{\pm0.05}$ |
| | MER | $-11.03_{\pm3.40}$ | $-2.18_{\pm3.51}$ | - | - | - | - |
| | GEM | $-10.26_{\pm3.08}$ | $-0.16_{\pm5.89}$ | $-7.50_{\pm7.05}$ | $0.13_{\pm3.54}$ | $0.42_{\pm0.35}$ | $54.06_{\pm4.35}$ |
| | A-GEM | $\mathbf{-10.04_{\pm3.11}}$ | $2.39_{\pm6.96}$ | $-11.37_{\pm0.08}$ | $-0.34_{\pm0.13}$ | $0.83_{\pm0.57}$ | $54.84_{\pm10.45}$ |
| | iCaRL | N/A | N/A | N/A | N/A | | - |
| | FDR | $-12.06_{\pm2.22}$ | $-0.81_{\pm3.89}$ | $-12.75_{\pm0.30}$ | $-2.42_{\pm0.86}$ | $-1.24_{\pm0.06}$ | $60.71_{\pm8.17}$ |
| | GSS | $-11.31_{\pm2.58}$ | $2.99_{\pm6.61}$ | $-7.08_{\pm10.01}$ | $\mathbf{6.17_{\pm2.06}}$ | $0.04_{\pm0.85}$ | $57.28_{\pm4.47}$ |
| | HAL | $-11.15_{\pm3.56}$ | $-0.20_{\pm3.99}$ | $-11.94_{\pm0.80}$ | $-0.02_{\pm0.10}$ | $\mathbf{1.72_{\pm0.08}}$ | $59.95_{\pm3.71}$ |
| | **DER (ours)** | $-10.16_{\pm3.78}$ | $\mathbf{3.23_{\pm5.24}}$ | $-11.89_{\pm0.88}$ | $0.27_{\pm7.12}$ | $1.23_{\pm0.26}$ | $64.69_{\pm2.02}$ |
| | **DER++ (ours)** | $-12.42_{\pm1.84}$ | $-2.33_{\pm5.69}$ | $\mathbf{-4.88_{\pm6.90}}$ | $2.68_{\pm0.11}$ | $0.91_{\pm0.45}$ | $\mathbf{67.21_{\pm2.13}}$ |
| 500 | ER | $-10.42_{\pm3.42}$ | $1.02_{\pm5.55}$ | $-8.42_{\pm4.83}$ | $-3.12_{\pm4.02}$ | $0.56_{\pm2.52}$ | $65.52_{\pm1.56}$ |
| | MER | $-10.59_{\pm3.83}$ | $0.89_{\pm5.03}$ | - | - | - | - |
| | GEM | $-10.59_{\pm3.26}$ | $0.11_{\pm5.66}$ | $-12.53_{\pm0.65}$ | $1.36_{\pm3.05}$ | $0.17_{\pm0.59}$ | $54.19_{\pm2.37}$ |
| | A-GEM | $-9.74_{\pm3.60}$ | $1.10_{\pm7.30}$ | $-6.38_{\pm8.64}$ | $\mathbf{6.36_{\pm3.88}}$ | $0.03_{\pm1.20}$ | $52.50_{\pm0.51}$ |
| | iCaRL | N/A | N/A | N/A | N/A | - | - |
| | FDR | $-9.27_{\pm2.80}$ | $4.73_{\pm5.08}$ | $\mathbf{-6.23_{\pm8.79}}$ | $3.71_{\pm2.70}$ | $-0.32_{\pm0.43}$ | $65.97_{\pm1.02}$ |
| | GSS | $-10.16_{\pm3.48}$ | $0.17_{\pm5.32}$ | $-7.84_{\pm4.43}$ | $2.11_{\pm3.31}$ | $\mathbf{0.89_{\pm0.94}}$ | $58.19_{\pm4.42}$ |
| | HAL | $-9.02_{\pm5.06}$ | $0.79_{\pm7.26}$ | $-7.15_{\pm7.57}$ | $3.06_{\pm1.03}$ | $1.33_{\pm0.23}$ | $64.21_{\pm3.16}$ |
| | **DER (ours)** | $\mathbf{-7.96_{\pm2.57}}$ | $1.17_{\pm6.37}$ | $-13.26_{\pm1.08}$ | $-4.52_{\pm2.39}$ | $0.21_{\pm1.21}$ | $\mathbf{72.45_{\pm0.14}}$ |
| | **DER++ (ours)** | $-10.90_{\pm4.88}$ | $-2.92_{\pm5.32}$ | $-6.29_{\pm8.89}$ | $-0.31_{\pm1.86}$ | $-0.35_{\pm0.01}$ | $67.05_{\pm0.11}$ |
| 5120 | ER | $-10.97_{\pm3.70}$ | $0.17_{\pm3.46}$ | $-8.45_{\pm10.75}$ | $-1.05_{\pm5.87}$ | $\mathbf{1.46_{\pm1.15}}$ | $\mathbf{73.03_{\pm1.59}}$ |
| | MER | $-10.50_{\pm3.35}$ | $-0.33_{\pm5.81}$ | - | - | - | - |
| | GEM | $-9.51_{\pm3.83}$ | $-0.28_{\pm9.16}$ | $-9.18_{\pm4.27}$ | $-1.24_{\pm0.83}$ | $1.03_{\pm0.89}$ | $62.06_{\pm3.01}$ |
| | A-GEM | $-11.31_{\pm3.44}$ | $\mathbf{1.14_{\pm7.08}}$ | $-8.01_{\pm6.31}$ | $-3.94_{\pm0.82}$ | $0.43_{\pm0.39}$ | $51.05_{\pm1.34}$ |
| | iCaRL | N/A | N/A | N/A | N/A | - | - |
| | FDR | $\mathbf{-9.25_{\pm4.65}}$ | $-1.30_{\pm5.90}$ | $-7.69_{\pm5.95}$ | $-0.52_{\pm0.54}$ | $-0.13_{\pm0.54}$ | $72.54_{\pm0.35}$ |
| | GSS | $-10.89_{\pm3.52}$ | $-2.19_{\pm6.64}$ | $-9.88_{\pm2.21}$ | $-0.13_{\pm5.24}$ | $0.34_{\pm1.49}$ | $63.39_{\pm4.55}$ |
| | HAL | $-10.06_{\pm4.46}$ | $0.16_{\pm7.43}$ | $-10.34_{\pm3.22}$ | $0.32_{\pm1.09}$ | $0.52_{\pm0.47}$ | $66.00_{\pm0.09}$ |
| | **DER (ours)** | $-11.59_{\pm4.34}$ | $-2.42_{\pm5.22}$ | $\mathbf{-5.98_{\pm8.44}}$ | $2.37_{\pm3.98}$ | $0.32_{\pm0.18}$ | $71.12_{\pm0.53}$ |
| | **DER++ (ours)** | $-10.71_{\pm2.95}$ | $0.20_{\pm9.44}$ | $-11.23_{\pm2.67}$ | $\mathbf{4.56_{\pm0.02}}$ | $0.06_{\pm0.22}$ | $72.11_{\pm1.81}$ |

Table 6: Forward Transfer results for the Experiments of Sec. 4.2 and F.1.

| | | BACKWARD TRANSFER | | | | | |
|---|---|---|---|---|---|---|---|
| | | S-MNIST | | S-CIFAR-10 | | P-MNIST | R-MNIST |
| **Buffer** | **Method** | *Class-IL* | *Task-IL* | *Class-IL* | *Task-IL* | *Domain-IL* | *Domain-IL* |
| – | SGD | $-99.10_{\pm0.55}$ | $-4.98_{\pm2.58}$ | $-96.39_{\pm0.12}$ | $-46.24_{\pm2.12}$ | $-57.65_{\pm4.32}$ | $-20.34_{\pm2.50}$ |
| | oEWC | $\mathbf{-97.79_{\pm1.24}}$ | $-0.38_{\pm0.19}$ | $\mathbf{-91.64_{\pm3.07}}$ | $-29.13_{\pm4.11}$ | $-36.69_{\pm2.34}$ | $-24.59_{\pm5.37}$ |
| | SI | $-98.89_{\pm0.86}$ | $-3.46_{\pm1.69}$ | $-95.78_{\pm0.64}$ | $-38.76_{\pm0.89}$ | $\mathbf{-27.91_{\pm0.31}}$ | $\mathbf{-22.91_{\pm0.26}}$ |
| – | LwF | $-99.30_{\pm0.11}$ | $-6.21_{\pm3.67}$ | $-96.69_{\pm0.25}$ | $-32.56_{\pm0.56}$ | - | - |
| | PNN | - | $\mathbf{0.00_{\pm0.00}}$ | - | $\mathbf{0.00_{\pm0.00}}$ | - | - |
| | ER | $-21.36_{\pm2.46}$ | $-0.82_{\pm0.41}$ | $-61.24_{\pm2.62}$ | $-7.08_{\pm0.64}$ | $-22.54_{\pm0.95}$ | $-8.24_{\pm1.56}$ |
| | MER | $-20.38_{\pm1.97}$ | $-0.81_{\pm0.20}$ | - | - | - | - |
| | GEM | $-22.32_{\pm2.04}$ | $-1.14_{\pm0.48}$ | $-82.61_{\pm1.60}$ | $-9.27_{\pm2.07}$ | $-29.38_{\pm2.56}$ | $-11.51_{\pm4.75}$ |
| | A-GEM | $-66.15_{\pm6.84}$ | $\mathbf{-0.06_{\pm2.95}}$ | $-95.73_{\pm0.20}$ | $-16.39_{\pm0.86}$ | $-31.69_{\pm3.92}$ | $-19.32_{\pm1.17}$ |
| 200 | iCaRL | $\mathbf{-11.73_{\pm0.73}}$ | $-0.23_{\pm0.06}$ | $\mathbf{-28.72_{\pm0.49}}$ | $\mathbf{-1.01_{\pm4.15}}$ | $-20.62_{\pm0.65}$ | $-13.31_{\pm2.60}$ |
| | FDR | $-21.15_{\pm4.18}$ | $-0.50_{\pm0.19}$ | $-86.40_{\pm2.67}$ | $-7.36_{\pm0.03}$ | | |
| | GSS | $-74.10_{\pm3.03}$ | $-4.29_{\pm2.31}$ | $-75.25_{\pm4.07}$ | $-8.56_{\pm1.78}$ | $-47.85_{\pm1.82}$ | $-20.19_{\pm6.45}$ |
| | HAL | $-14.54_{\pm1.49}$ | $-0.48_{\pm0.20}$ | $-69.11_{\pm4.21}$ | $-11.91_{\pm0.52}$ | $-15.24_{\pm1.33}$ | $-11.71_{\pm0.26}$ |
| | **DER (ours)** | $-17.66_{\pm2.10}$ | $-0.56_{\pm0.18}$ | $-40.76_{\pm0.42}$ | $-6.21_{\pm0.71}$ | $-13.79_{\pm0.80}$ | $-5.99_{\pm0.46}$ |
| | **DER++ (ours)** | $-16.27_{\pm1.73}$ | $-0.55_{\pm0.37}$ | $-32.59_{\pm2.32}$ | $-5.16_{\pm0.21}$ | $\mathbf{-11.47_{\pm0.33}}$ | $\mathbf{-5.27_{\pm0.26}}$ |
| | ER | $-15.97_{\pm2.46}$ | $-0.36_{\pm0.20}$ | $-45.35_{\pm0.07}$ | $-3.54_{\pm0.35}$ | $-14.90_{\pm0.39}$ | $-7.52_{\pm1.44}$ |
| | MER | $-11.52_{\pm0.56}$ | $-0.44_{\pm0.17}$ | - | - | - | - |
| | GEM | $-15.47_{\pm2.03}$ | $-0.27_{\pm0.98}$ | $-74.31_{\pm4.62}$ | $-9.12_{\pm0.21}$ | $-18.76_{\pm0.91}$ | $-7.19_{\pm1.40}$ |
| | A-GEM | $-65.84_{\pm7.24}$ | $-0.54_{\pm0.20}$ | $-94.01_{\pm1.16}$ | $-14.26_{\pm4.18}$ | $-28.53_{\pm2.01}$ | $-19.36_{\pm3.18}$ |
| 500 | iCaRL | $-11.84_{\pm0.73}$ | $\mathbf{-0.25_{\pm0.09}}$ | $-25.71_{\pm1.10}$ | $\mathbf{-1.06_{\pm4.21}}$ | - | - |
| | FDR | $-13.90_{\pm5.19}$ | $-1.27_{\pm2.43}$ | $-85.62_{\pm0.36}$ | $-4.80_{\pm0.30}$ | $-12.80_{\pm1.28}$ | $-6.70_{\pm1.93}$ |
| | GSS | $-60.35_{\pm6.03}$ | $-0.77_{\pm0.62}$ | $-62.88_{\pm2.67}$ | $-7.73_{\pm3.99}$ | $-23.68_{\pm1.35}$ | $-17.45_{\pm9.92}$ |
| | HAL | $-9.97_{\pm1.62}$ | $-0.30_{\pm0.26}$ | $-62.21_{\pm4.34}$ | $-5.41_{\pm1.10}$ | $-11.58_{\pm0.49}$ | $-6.78_{\pm0.87}$ |
| | **DER (ours)** | $-9.58_{\pm1.52}$ | $-0.39_{\pm0.18}$ | $-26.74_{\pm0.15}$ | $-4.56_{\pm0.45}$ | $-8.04_{\pm0.42}$ | $-3.41_{\pm2.18}$ |
| | **DER++ (ours)** | $\mathbf{-8.85_{\pm1.86}}$ | $-0.34_{\pm0.16}$ | $\mathbf{-22.38_{\pm4.41}}$ | $-4.66_{\pm1.15}$ | $\mathbf{-7.62_{\pm1.02}}$ | $\mathbf{-3.18_{\pm0.14}}$ |
| | ER | $-6.07_{\pm1.84}$ | $0.03_{\pm0.36}$ | $-13.99_{\pm1.12}$ | $\mathbf{0.08_{\pm0.06}}$ | $-5.24_{\pm0.13}$ | $-2.55_{\pm0.53}$ |
| | MER | $\mathbf{-3.22_{\pm0.33}}$ | $0.05_{\pm0.11}$ | - | - | - | - |
| | GEM | $-4.14_{\pm1.43}$ | $0.16_{\pm0.85}$ | $-75.27_{\pm4.41}$ | $-6.91_{\pm2.33}$ | $-6.74_{\pm0.49}$ | $\mathbf{-0.06_{\pm0.29}}$ |
| | A-GEM | $-55.04_{\pm10.93}$ | $\mathbf{0.78_{\pm4.16}}$ | $-84.49_{\pm3.08}$ | $-9.89_{\pm0.40}$ | $-23.73_{\pm2.22}$ | $-17.70_{\pm1.28}$ |
| 5120 | iCaRL | $-11.64_{\pm0.72}$ | $-0.22_{\pm0.08}$ | $-24.94_{\pm0.14}$ | $-0.99_{\pm1.41}$ | - | - |
| | FDR | $-11.58_{\pm3.97}$ | $-0.87_{\pm1.66}$ | $-96.64_{\pm0.19}$ | $-1.89_{\pm0.51}$ | $-3.81_{\pm0.13}$ | $-2.81_{\pm0.47}$ |
| | GSS | $-7.90_{\pm1.21}$ | $-0.09_{\pm0.15}$ | $-58.11_{\pm9.12}$ | $-6.38_{\pm1.71}$ | $-19.82_{\pm1.31}$ | $-17.05_{\pm2.31}$ |
| | HAL | $-6.55_{\pm1.63}$ | $0.02_{\pm0.20}$ | $-27.19_{\pm7.53}$ | $-4.51_{\pm0.54}$ | $-4.27_{\pm0.22}$ | $-2.25_{\pm0.01}$ |
| | **DER (ours)** | $-4.53_{\pm0.83}$ | $-0.31_{\pm0.08}$ | $-10.12_{\pm0.80}$ | $-2.59_{\pm0.08}$ | $-3.49_{\pm0.02}$ | $-1.73_{\pm0.10}$ |
| | **DER++ (ours)** | $-4.19_{\pm1.63}$ | $-0.13_{\pm0.09}$ | $\mathbf{-6.89_{\pm0.50}}$ | $-1.16_{\pm0.22}$ | $\mathbf{-2.93_{\pm0.15}}$ | $-1.18_{\pm0.53}$ |

Table 7: Backward Transfer results for the Experiments of Sec. 4.2 and F.1.

| | | FORGETTING | | | | | |
|---|---|---|---|---|---|---|---|
| | | S-MNIST | | S-CIFAR-10 | | P-MNIST | R-MNIST |
| Buffer | Method | Class-IL | Task-IL | Class-IL | Task-IL | Domain-IL | Domain-IL |
| – | SGD | $99.10_{\pm0.55}$ | $5.15_{\pm2.74}$ | $96.39_{\pm0.12}$ | $46.24_{\pm2.12}$ | $57.65_{\pm4.32}$ | $20.82_{\pm2.47}$ |
| – | oEWC | $\mathbf{97.79_{\pm1.24}}$ | $0.44_{\pm0.16}$ | $\mathbf{91.64_{\pm3.07}}$ | $29.33_{\pm3.84}$ | $36.69_{\pm2.34}$ | $36.44_{\pm1.44}$ |
| | SI | $98.89_{\pm0.86}$ | $5.15_{\pm2.74}$ | $95.78_{\pm0.64}$ | $38.76_{\pm0.89}$ | $\mathbf{27.91_{\pm0.31}}$ | $\mathbf{23.41_{\pm0.49}}$ |
| | LwF | $99.30_{\pm0.11}$ | $5.15_{\pm2.74}$ | $96.69_{\pm0.25}$ | $32.56_{\pm0.56}$ | - | - |
| | PNN | - | $\mathbf{0.00_{\pm0.00}}$ | - | $\mathbf{0.00_{\pm0.00}}$ | - | - |
| 200 | ER | $21.36_{\pm2.46}$ | $0.84_{\pm0.41}$ | $61.24_{\pm2.62}$ | $7.08_{\pm0.64}$ | $22.54_{\pm0.95}$ | $8.87_{\pm1.44}$ |
| | MER | $20.38_{\pm1.97}$ | $0.82_{\pm0.21}$ | - | - | - | - |
| | GEM | $22.32_{\pm2.04}$ | $1.19_{\pm0.38}$ | $82.61_{\pm1.60}$ | $9.27_{\pm2.07}$ | $29.38_{\pm2.56}$ | $12.97_{\pm4.82}$ |
| | A-GEM | $66.15_{\pm6.84}$ | $0.96_{\pm0.28}$ | $95.73_{\pm0.20}$ | $16.39_{\pm0.86}$ | $31.69_{\pm3.92}$ | $20.05_{\pm1.12}$ |
| | iCaRL | $\mathbf{11.73_{\pm0.73}}$ | $\mathbf{0.28_{\pm0.08}}$ | $\mathbf{28.72_{\pm0.49}}$ | $\mathbf{2.63_{\pm3.48}}$ | - | - |
| | FDR | $21.15_{\pm4.18}$ | $0.52_{\pm0.18}$ | $86.40_{\pm2.67}$ | $7.36_{\pm0.03}$ | $20.62_{\pm0.65}$ | $13.66_{\pm2.52}$ |
| | GSS | $74.10_{\pm3.03}$ | $4.30_{\pm2.31}$ | $75.25_{\pm4.07}$ | $8.56_{\pm1.78}$ | $47.85_{\pm1.82}$ | $20.71_{\pm6.50}$ |
| | HAL | $14.54_{\pm1.49}$ | $0.53_{\pm0.19}$ | $69.11_{\pm4.21}$ | $12.26_{\pm0.02}$ | $79.00_{\pm1.17}$ | $83.59_{\pm0.04}$ |
| | **DER (ours)** | $17.66_{\pm2.10}$ | $0.57_{\pm0.18}$ | $40.76_{\pm0.42}$ | $6.57_{\pm0.20}$ | $14.00_{\pm0.73}$ | $6.53_{\pm0.32}$ |
| | **DER++ (ours)** | $16.27_{\pm1.73}$ | $0.66_{\pm0.28}$ | $32.59_{\pm2.32}$ | $5.16_{\pm0.21}$ | $\mathbf{11.49_{\pm0.31}}$ | $\mathbf{6.08_{\pm0.43}}$ |
| 500 | ER | $15.97_{\pm2.46}$ | $0.39_{\pm0.20}$ | $45.35_{\pm0.07}$ | $3.54_{\pm0.35}$ | $14.90_{\pm0.39}$ | $8.02_{\pm1.56}$ |
| | MER | $11.52_{\pm0.56}$ | $0.45_{\pm0.17}$ | - | - | - | - |
| | GEM | $15.57_{\pm1.77}$ | $0.54_{\pm0.15}$ | $74.31_{\pm4.62}$ | $9.12_{\pm0.21}$ | $18.76_{\pm0.91}$ | $8.79_{\pm1.44}$ |
| | A-GEM | $65.84_{\pm7.24}$ | $0.64_{\pm0.20}$ | $94.01_{\pm1.16}$ | $14.26_{\pm4.18}$ | $28.53_{\pm2.01}$ | $19.70_{\pm3.14}$ |
| | iCaRL | $11.84_{\pm0.73}$ | $\mathbf{0.30_{\pm0.09}}$ | $25.71_{\pm1.10}$ | $\mathbf{2.66_{\pm2.47}}$ | - | - |
| | FDR | $13.90_{\pm5.19}$ | $1.35_{\pm2.40}$ | $85.62_{\pm0.36}$ | $4.80_{\pm0.00}$ | $12.80_{\pm1.28}$ | $7.21_{\pm1.89}$ |
| | GSS | $60.35_{\pm6.03}$ | $0.89_{\pm0.40}$ | $62.88_{\pm2.67}$ | $7.73_{\pm3.99}$ | $23.68_{\pm1.35}$ | $18.05_{\pm9.89}$ |
| | HAL | $9.97_{\pm1.62}$ | $0.35_{\pm0.21}$ | $62.21_{\pm4.34}$ | $5.41_{\pm1.10}$ | $82.53_{\pm0.36}$ | $88.53_{\pm0.77}$ |
| | **DER (ours)** | $9.58_{\pm1.52}$ | $0.45_{\pm0.13}$ | $26.74_{\pm0.15}$ | $4.56_{\pm0.45}$ | $8.07_{\pm0.43}$ | $3.96_{\pm2.08}$ |
| | **DER++ (ours)** | $\mathbf{8.85_{\pm1.86}}$ | $0.35_{\pm0.15}$ | $\mathbf{22.38_{\pm4.41}}$ | $4.66_{\pm1.15}$ | $\mathbf{7.67_{\pm1.05}}$ | $\mathbf{3.57_{\pm0.09}}$ |
| 5120 | ER | $6.08_{\pm1.84}$ | $0.25_{\pm0.23}$ | $13.99_{\pm1.12}$ | $0.27_{\pm0.06}$ | $5.24_{\pm0.13}$ | $3.10_{\pm0.42}$ |
| | MER | $\mathbf{3.22_{\pm0.33}}$ | $\mathbf{0.07_{\pm0.06}}$ | - | - | - | - |
| | GEM | $4.30_{\pm1.16}$ | $0.16_{\pm0.09}$ | $75.27_{\pm4.41}$ | $6.91_{\pm2.33}$ | $6.74_{\pm0.49}$ | $2.49_{\pm0.17}$ |
| | A-GEM | $55.10_{\pm10.79}$ | $0.63_{\pm0.21}$ | $84.49_{\pm3.08}$ | $11.36_{\pm1.68}$ | $23.74_{\pm2.23}$ | $18.10_{\pm1.44}$ |
| | iCaRL | $11.64_{\pm0.72}$ | $0.26_{\pm0.06}$ | $24.94_{\pm0.14}$ | $1.59_{\pm0.57}$ | - | - |
| | FDR | $11.58_{\pm3.97}$ | $0.95_{\pm1.61}$ | $96.64_{\pm0.19}$ | $1.93_{\pm0.48}$ | $3.82_{\pm0.12}$ | $3.31_{\pm0.56}$ |
| | GSS | $7.90_{\pm1.21}$ | $0.18_{\pm0.11}$ | $58.11_{\pm9.12}$ | $7.71_{\pm2.31}$ | $89.76_{\pm0.39}$ | $92.66_{\pm0.02}$ |
| | HAL | $6.55_{\pm1.63}$ | $0.13_{\pm0.07}$ | $27.19_{\pm7.53}$ | $5.21_{\pm0.50}$ | $19.97_{\pm1.33}$ | $17.62_{\pm2.33}$ |
| | **DER (ours)** | $4.53_{\pm0.83}$ | $0.32_{\pm0.08}$ | $10.12_{\pm0.80}$ | $2.59_{\pm0.08}$ | $3.51_{\pm0.03}$ | $2.17_{\pm0.11}$ |
| | **DER++ (ours)** | $4.19_{\pm1.63}$ | $0.23_{\pm0.06}$ | $\mathbf{7.27_{\pm0.84}}$ | $\mathbf{1.18_{\pm0.19}}$ | $\mathbf{2.96_{\pm0.14}}$ | $\mathbf{1.62_{\pm0.50}}$ |

Table 8: Forgetting results for the Experiments of Sec. 4.2 and F.1.

# G   Hyperparameter Search

## G.1   Best values

In Table 9, we show the best hyperparameter combination that we chose for each method for the experiments in the main paper, according to the criteria outlined in Section 4.1. We denote the learning rate with *lr*, the batch size with *bs* and the minibatch size (i.e. the size of the batches drawn from the buffer in rehearsal-based methods) with *mbs*, while other symbols refer to the respective methods. We hold *batch size* and *minibatch size* out of the hyperparameter search space for all Continual Learning benchmarks. Their values are fixed as follows: Sequential MNIST: 10; Sequential CIFAR-10, Sequential Tiny ImageNet: 32; Permuted MNIST, Rotated MNIST: 128.

Conversely, *batch size* and *minibatch size* belong to the hyperparameter search space for experiments on the novel MNIST-360 dataset. It must be noted that MER does not depend on *batch size*, as it internally always adopts a single-example forward pass.

| Method | Buffer | Permuted MNIST | Buffer | Rotated MNIST |
|---|---|---|---|---|
| SGD | – | *lr:* 0.2 | – | *lr:* 0.2 |
| oEWC | – | *lr:* 0.1   *λ:* 0.7   *γ:* 1.0 | – | *lr:* 0.1   *λ:* 0.7   *γ:* 1.0 |
| SI | – | *lr:* 0.1   *c:* 0.5   *ξ:* 1.0 | – | *lr:* 0.1   *c:* 1.0   *ξ:* 1.0 |
| ER | 200 | *lr:* 0.2 | 200 | *lr:* 0.2 |
|  | 500 | *lr:* 0.2 | 500 | *lr:* 0.2 |
|  | 5120 | *lr:* 0.2 | 5120 | *lr:* 0.2 |
| GEM | 200 | *lr:* 0.1   *γ:* 0.5 | 200 | *lr:* 0.01   *γ:* 0.5 |
|  | 500 | *lr:* 0.1   *γ:* 0.5 | 500 | *lr:* 0.01   *γ:* 0.5 |
|  | 5120 | *lr:* 0.1   *γ:* 0.5 | 5120 | *lr:* 0.01   *γ:* 0.5 |
| A-GEM | 200 | *lr:* 0.1 | 200 | *lr:* 0.1 |
|  | 500 | *lr:* 0.1 | 500 | *lr:* 0.3 |
|  | 5120 | *lr:* 0.1 | 5120 | *lr:* 0.3 |
| FDR | 200 | *lr:* 0.1   *α:* 1.0 | 200 | *lr:* 0.1   *α:* 1.0 |
|  | 500 | *lr:* 0.1   *α:* 0.3 | 500 | *lr:* 0.2   *α:* 0.3 |
|  | 5120 | *lr:* 0.1   *α:* 1.0 | 5120 | *lr:* 0.2   *α:* 1.0 |
| GSS | 200 | *lr:* 0.2   *gmbs:* 128   *nb:* 1 | 200 | *lr:* 0.2   *gmbs:* 128   *nb:* 1 |
|  | 500 | *lr:* 0.1   *gmbs:* 10   *nb:* 1 | 500 | *lr:* 0.2   *gmbs:* 128   *nb:* 1 |
|  | 5120 | *lr:* 0.03   *gmbs:* 10   *nb:* 1 | 5120 | *lr:* 0.2   *gmbs:* 128   *nb:* 1 |
| HAL | 200 | *lr:* 0.1   *λ:* 0.1   *β:* 0.5   *γ:* 0.1 | 200 | *lr:* 0.1   *λ:* 0.2   *β:* 0.5   *γ:* 0.1 |
|  | 500 | *lr:* 0.1   *λ:* 0.1   *β:* 0.3   *γ:* 0.1 | 500 | *lr:* 0.1   *λ:* 0.1   *β:* 0.5   *γ:* 0.1 |
|  | 5120 | *lr:* 0.1   *λ:* 0.1   *β:* 0.5   *γ:* 0.1 | 5120 | *lr:* 0.1   *λ:* 0.1   *β:* 0.3   *γ:* 0.1 |
| DER | 200 | *lr:* 0.2   *α:* 1.0 | 200 | *lr:* 0.2   *α:* 1.0 |
|  | 500 | *lr:* 0.2   *α:* 1.0 | 500 | *lr:* 0.2   *α:* 0.5 |
|  | 5120 | *lr:* 0.2   *α:* 0.5 | 5120 | *lr:* 0.2   *α:* 0.5 |
| DER++ | 200 | *lr:* 0.1   *α:* 1.0   *β:* 1.0 | 200 | *lr:* 0.1   *α:* 1.0   *β:* 0.5 |
|  | 500 | *lr:* 0.2   *α:* 1.0   *β:* 0.5 | 500 | *lr:* 0.2   *α:* 0.5   *β:* 1.0 |
|  | 5120 | *lr:* 0.2   *α:* 0.5   *β:* 1.0 | 5120 | *lr:* 0.2   *α:* 0.5   *β:* 0.5 |

| Method | Buffer | Sequential MNIST | Buffer | Sequential CIFAR-10 |
|---|---|---|---|---|
| SGD | – | *lr:* 0.03 | – | *lr:* 0.1 |
| oEWC | – | *lr:* 0.03   *λ:* 90   *γ:* 1.0 | – | *lr:* 0.03   *λ:* 10   *γ:* 1.0 |
| SI | – | *lr:* 0.1   *c:* 1.0   *ξ:* 0.9 | – | *lr:* 0.03   *c:* 0.5   *ξ:* 1.0 |
| LwF | – | *lr:* 0.03   *α:* 1   *T:* 2.0   *wd:* 0.0005 | – | *lr:* 0.03   *α:* 0.5   *T:* 2.0 |
| PNN | – | *lr:* 0.1 | – | *lr:* 0.03 |
| ER | 200 | *lr:* 0.01 | 200 | *lr:* 0.1 |
|  | 500 | *lr:* 0.1 | 500 | *lr:* 0.1 |
|  | 5120 | *lr:* 0.1 | 5120 | *lr:* 0.1 |
| MER | 200 | *lr:* 0.1   *β:* 1   *γ:* 1   *nb:* 1   *bs:* 1 |  |  |
|  | 500 | *lr:* 0.1   *β:* 1   *γ:* 1   *nb:* 1   *bs:* 1 |  |  |
|  | 5120 | *lr:* 0.03   *β:* 1   *γ:* 1   *nb:* 1   *bs:* 1 |  |  |
| GEM | 200 | *lr:* 0.01   *γ:* 1.0 | 200 | *lr:* 0.03   *γ:* 0.5 |
|  | 500 | *lr:* 0.03   *γ:* 0.5 | 500 | *lr:* 0.03   *γ:* 0.5 |
|  | 5120 | *lr:* 0.1   *γ:* 1.0 | 5120 | *lr:* 0.03   *γ:* 0.5 |

| Method | Buffer | Sequential MNIST | Buffer | Sequential CIFAR-10 |
|--------|--------|------------------|--------|---------------------|
| A-GEM | 200 | *lr:* 0.1 | 200 | *lr:* 0.03 |
| | 500 | *lr:* 0.1 | 500 | *lr:* 0.03 |
| | 5120 | *lr:* 0.1 | 5120 | *lr:* 0.03 |
| iCaRL | 200 | *lr:* 0.1  *wd:* 0 | 200 | *lr:* 0.1  *wd:* $10^{-5}$ |
| | 500 | *lr:* 0.1  *wd:* 0 | 500 | *lr:* 0.1  *wd:* $10^{-5}$ |
| | 5120 | *lr:* 0.1  *wd:* 0 | 5120 | *lr:* 0.03  *wd:* $10^{-5}$ |
| FDR | 200 | *lr:* 0.03  $\alpha$: 0.5 | 200 | *lr:* 0.03  $\alpha$: 0.3 |
| | 500 | *lr:* 0.1  $\alpha$: 0.2 | 500 | *lr:* 0.03  $\alpha$: 1.0 |
| | 5120 | *lr:* 0.1  $\alpha$: 0.2 | 5120 | *lr:* 0.03  $\alpha$: 0.3 |
| GSS | 200 | *lr:* 0.1  *gmbs:* 10  *nb:* 1 | 200 | *lr:* 0.03  *gmbs:* 32  *nb:* 1 |
| | 500 | *lr:* 0.1  *gmbs:* 10  *nb:* 1 | 500 | *lr:* 0.03  *gmbs:* 32  *nb:* 1 |
| | 5120 | *lr:* 0.1  *gmbs:* 10  *nb:* 1 | 5120 | *lr:* 0.03  *gmbs:* 32  *nb:* 1 |
| HAL | 200 | *lr:* 0.1  $\lambda$: 0.1  $\beta$: 0.7  $\gamma$: 0.5 | 200 | *lr:* 0.03  $\lambda$: 0.2  $\beta$: 0.5  $\gamma$: 0.1 |
| | 500 | *lr:* 0.1  $\lambda$: 0.1  $\beta$: 0.2  $\gamma$: 0.5 | 500 | *lr:* 0.03  $\lambda$: 0.1  $\beta$: 0.3  $\gamma$: 0.1 |
| | 5120 | *lr:* 0.1  $\lambda$: 0.1  $\beta$: 0.7  $\gamma$: 0.5 | 5120 | *lr:* 0.03  $\lambda$: 0.1  $\beta$: 0.3  $\gamma$: 0.1 |
| DER | 200 | *lr:* 0.03  $\alpha$: 0.2 | 200 | *lr:* 0.03  $\alpha$: 0.3 |
| | 500 | *lr:* 0.03  $\alpha$: 1.0 | 500 | *lr:* 0.03  $\alpha$: 0.3 |
| | 5120 | *lr:* 0.1  $\alpha$: 0.5 | 5120 | *lr:* 0.03  $\alpha$: 0.3 |
| DER++ | 200 | *lr:* 0.03  $\alpha$: 0.2  $\beta$: 1.0 | 200 | *lr:* 0.03  $\alpha$: 0.1  $\beta$: 0.5 |
| | 500 | *lr:* 0.03  $\alpha$: 1.0  $\beta$: 0.5 | 500 | *lr:* 0.03  $\alpha$: 0.2  $\beta$: 0.5 |
| | 5120 | *lr:* 0.1  $\alpha$: 0.2  $\beta$: 0.5 | 5120 | *lr:* 0.03  $\alpha$: 0.1  $\beta$: 1.0 |

| Method | Buffer | Sequential Tiny ImageNet | Buffer | MNIST-360 |
|--------|--------|--------------------------|--------|-----------|
| SGD | – | *lr:* 0.03 | – | *lr:* 0.1  *bs:* 4 |
| oEWC | – | *lr:* 0.03  $\lambda$: 25  $\gamma$: 1.0 | | |
| SI | – | *lr:* 0.03  *c:* 0.5  $\xi$: 1.0 | | |
| LwF | – | *lr:* 0.01  $\alpha$: 1.0  *T:* 2.0 | | |
| PNN | – | *lr:* 0.03 | | |
| ER | 200 | *lr:* 0.1 | 200 | *lr:* 0.2  *bs:* 1  *mbs:* 16 |
| | 500 | *lr:* 0.03 | 500 | *lr:* 0.2  *bs:* 1  *mbs:* 16 |
| | 5120 | *lr:* 0.1 | 1000 | *lr:* 0.2  *bs:* 4  *mbs:* 16 |
| MER | | | 200 | *lr:* 0.2  *mbs:* 128  $\beta$: 1  $\gamma$: 1  *nb:* 3 |
| | | | 500 | *lr:* 0.1  *mbs:* 128  $\beta$: 1  $\gamma$: 1  *nb:* 3 |
| | | | 1000 | *lr:* 0.2  *mbs:* 128  $\beta$: 1  $\gamma$: 1  *nb:* 3 |
| A-GEM | 200 | *lr:* 0.01 | 200 | *lr:* 0.1  *bs:* 16  *mbs:* 128 |
| | 500 | *lr:* 0.01 | 500 | *lr:* 0.1  *bs:* 16  *mbs:* 128 |
| | 5120 | *lr:* 0.01 | 1000 | *lr:* 0.1  *bs:* 4  *mbs:* 128 |
| iCaRL | 200 | *lr:* 0.03  *wd:* $10^{-5}$ | | |
| | 500 | *lr:* 0.03  *wd:* $10^{-5}$ | | |
| | 5120 | *lr:* 0.03  *wd:* $10^{-5}$ | | |

| Method | Buffer | Sequential Tiny Imagenet | Buffer | MNIST-360 |
|---|---|---|---|---|
| FDR | 200 | *lr:* 0.03 *α:* 0.3 | | |
| | 500 | *lr:* 0.03 *α:* 1.0 | | |
| | 5120 | *lr:* 0.03 *α:* 0.3 | | |
| GSS | | | 200 | *lr:* 0.2 *bs:* 1 *mbs:* 16 |
| | | | 500 | *lr:* 0.2 *bs:* 1 *mbs:* 16 |
| | | | 1000 | *lr:* 0.2 *bs:* 4 *mbs:* 16 |
| DER | 200 | *lr:* 0.03 *α:* 0.1 | 200 | *lr:* 0.1 *bs:* 16 *mbs:* 64 *α:* 0.5 |
| | 500 | *lr:* 0.03 *α:* 0.1 | 500 | *lr:* 0.2 *bs:* 16 *mbs:* 16 *α:* 0.5 |
| | 5120 | *lr:* 0.03 *α:* 0.1 | 1000 | *lr:* 0.1 *bs:* 8 *mbs:* 16 *α:* 0.5 |
| DER++ | 200 | *lr:* 0.03 *α:* 0.1 *β:* 1.0 | 200 | *lr:* 0.2 *bs:* 16 *mbs:* 16 *α:* 0.5 *β:* 1.0 |
| | 500 | *lr:* 0.03 *α:* 0.2 *β:* 0.5 | 500 | *lr:* 0.2 *bs:* 16 *mbs:* 16 *α:* 0.5 *β:* 1.0 |
| | 5120 | *lr:* 0.03 *α:* 0.1 *β:* 0.5 | 1000 | *lr:* 0.2 *bs:* 16 *mbs:* 128 *α:* 0.2 *β:* 1.0 |

Table 9: Hyperparameters selected for our experiments.

## G.2 All values

In the following, we provide a list of all the parameter combinations that were considered (Table 10). Note that the same parameters are searched for all examined buffer sizes.

**Permuted MNIST**

| Method | Par | Values |
|---|---|---|
| SGD/JOINT | *lr* | [0.03, 0.1, 0.2] |
| oEWC | *lr* | [0.1, 0.01] |
| | $\lambda$ | [0.1, 1, 10, 30, 90,100] |
| | $\gamma$ | [0.9, 1.0] |
| SI | *lr* | [0.01, 0.1] |
| | *c* | [0.5, 1.0] |
| | $\xi$ | [0.9, 1.0] |
| ER | *lr* | [0.03, 0.1, 0.2] |
| GEM | *lr* | [0.01, 0.1, 0.3] |
| | $\gamma$ | [0.1, 0.5, 1] |
| A-GEM | *lr* | [0.01, 0.1, 0.3] |
| GSS | *lr* | [0.03, 0.1, 0.2] |
| | *gmbs* | [10, 64, 128] |
| | *nb* | [1] |
| HAL | *lr* | [0.03, 0.1, 0.3] |
| | $\lambda$ | [0.1, 0.2] |
| | $\beta$ | [0.3, 0.5] |
| | $\gamma$ | [0.1] |
| DER | *lr* | [0.1, 0.2] |
| | $\alpha$ | [0.5, 1.0] |
| DER++ | *lr* | [0.1, 0.2] |
| | $\alpha$ | [0.5, 1.0] |
| | $\beta$ | [0.5, 1.0] |

**Rotated MNIST**

| Method | Par | Values |
|---|---|---|
| SGD | *lr* | [0.03, 0.1, 0.2] |
| oEWC | *lr* | [0.01, 0.1] |
| | $\lambda$ | [0.1, 0.7, 1, 10, 30, 90,100] |
| | $\gamma$ | [0.9, 1.0] |
| SI | *lr* | [0.01, 0.1] |
| | *c* | [0.5, 1.0] |
| | $\xi$ | [0.9, 1.0] |
| ER | *lr* | [0.1, 0.2] |
| GEM | *lr* | [0.01, 0.3, 0.1] |
| | $\gamma$ | [0.1, 0.5, 1] |
| A-GEM | *lr* | [0.01, 0.1, 0.3] |
| GSS | *lr* | [0.03, 0.1, 0.2] |
| | *gmbs* | [10, 64, 128] |
| | *nb* | [1] |
| HAL | *lr* | [0.03, 0.1, 0.3] |
| | $\lambda$ | [0.1, 0.2] |
| | $\beta$ | [0.3, 0.5] |
| | $\gamma$ | [0.1] |
| DER | *lr* | [0.1, 0.2] |
| | $\alpha$ | [0.5, 1.0] |
| DER++ | *lr* | [0.1, 0.2] |
| | $\alpha$ | [0.5, 1.0] |
| | $\beta$ | [0.5, 1.0] |

**Sequential Tiny ImageNet**

| Method | Par | Values |
|---|---|---|
| SGD | *lr* | [0.01, 0.03, 0.1] |
| oEWC | *lr* | [0.01, 0.03] |
| | $\lambda$ | [10, 25, 30, 90, 100] |
| | $\gamma$ | [0.9, 0.95, 1.0] |
| SI | *lr* | [0.01, 0.03] |
| | *c* | [0.5] |
| | $\xi$ | [1.0] |
| LwF | *lr* | [0.01, 0.03] |
| | $\alpha$ | [0.3, 1, 3] |
| | *T* | [2.0, 4.0] |
| | *wd* | [0.00005, 0.00001] |
| PNN | *lr* | [0.03, 0.1] |
| ER | *lr* | [0.01, 0.03, 0.1] |
| A-GEM | *lr* | [0.003, 0.01] |
| iCaRL | *lr* | [0.01, 0.03, 0.1] |
| | *wd* | [0.00005, 0.00001] |
| FDR | *lr* | [0.01, 0.03, 0.1] |
| | $\alpha$ | [0.03, 0.1, 0.3, 1.0, 3.0] |
| DER | *lr* | [0.01, 0.03, 0.1] |
| | $\alpha$ | [0.1, 0.5, 1.0] |
| DER++ | *lr* | [0.01, 0.03] |
| | $\alpha$ | [0.1, 0.2, 0.5, 1.0] |
| | $\beta$ | [0.5, 1.0] |

| Sequential MNIST | | |
|---|---|---|
| *Method* | *Par* | *Values* |
| SGD | $lr$ | [0.01, 0.03, 0.1] |
| oEWC | $lr$ | [0.01, 0.03, 0.1] |
| | $\lambda$ | [10, 25, 30, 90, 100] |
| | $\gamma$ | [0.9, 1.0] |
| SI | $lr$ | [0.01, 0.03, 0.1] |
| | $c$ | [0.3, 0.5, 0.7, 1.0] |
| | $\xi$ | [0.9, 1.0] |
| LwF | $lr$ | [0.01, 0.03, 0.1] |
| | $\alpha$ | [0.3, 0.5, 0.7, 1.0] |
| | $T$ | [2.0, 4.0] |
| PNN | $lr$ | [0.01, 0.03, 0.1] |
| ER | $lr$ | [0.03, 0.01, 0.1] |
| MER | $lr$ | [0.03, 0.1] |
| | $\beta, \gamma$ | [1.0] |
| | $nb$ | [1] |
| GEM | $lr$ | [0.01, 0.03, 0.1] |
| | $\gamma$ | [0.5, 1] |
| A-GEM | $lr$ | [0.03, 0.1] |
| iCaRL | $lr$ | [0.03, 0.1] |
| | $wd$ | [0.0001, 0.0005] |
| FDR | $lr$ | [0.03, 0.1] |
| | $\alpha$ | [0.2, 0.5, 1.0] |
| GSS | $lr$ | [0.03, 0.1] |
| | $gmbs$ | [10] |
| | $nb$ | [1] |
| HAL | $lr$ | [0.03, 0.1, 0.2] |
| | $\lambda$ | [0.1, 0.5] |
| | $\beta$ | [0.2, 0.5, 0.7] |
| | $\gamma$ | [0.1, 0.5] |
| DER | $lr$ | [0.03, 0.1] |
| | $\alpha$ | [0.2, 0.5, 1.0] |
| DER++ | $lr$ | [0.03, 0.1] |
| | $\alpha$ | [0.2, 0.5, 1.0] |
| | $\beta$ | [0.2, 0.5, 1.0] |

| Sequential CIFAR-10 | | |
|---|---|---|
| *Method* | *Par* | *Values* |
| SGD | $lr$ | [0.01, 0.03, 0.1] |
| oEWC | $lr$ | [0.03, 0.1] |
| | $\lambda$ | [10, 25, 30, 90, 100] |
| | $\gamma$ | [0.9, 1.0] |
| SI | $lr$ | [0.01, 0.03, 0.1] |
| | $c$ | [0.5, 0.1] |
| | $\xi$ | [1.0] |
| LwF | $lr$ | [0.01, 0.03, 0.1] |
| | $\alpha$ | [0.3, 1, 3, 10] |
| | $T$ | [2.0] |
| PNN | $lr$ | [0.03, 0.1] |
| ER | $lr$ | [0.01, 0.03, 0.1] |
| GEM | $lr$ | [0.01, 0.03, 0.1] |
| | $\gamma$ | [0.5, 1] |
| A-GEM | $lr$ | [0.03, 0.1] |
| iCaRL | $lr$ | [0.01, 0.03, 0.1] |
| | $wd$ | [0.0001, 0.0005] |
| FDR | $lr$ | [0.01, 0.03, 0.1] |
| | $\alpha$ | [0.03, 0.1, 0.3, 1.0, 3.0] |
| GSS | $lr$ | [0.01, 0.03, 0.1] |
| | $gmbs$ | [32] |
| | $nb$ | [1] |
| HAL | $lr$ | [0.01, 0.03, 0.1] |
| | $\lambda$ | [0.1, 0.5] |
| | $\beta$ | [0.2, 0.3, 0.5] |
| | $\gamma$ | [0.1] |
| DER | $lr$ | [0.01, 0.03, 0.1] |
| | $\alpha$ | [0.2, 0.5, 1.0] |
| DER++ | $lr$ | [0.01, 0.03, 0.1] |
| | $\alpha$ | [0.1, 0.2, 0.5] |
| | $\beta$ | [0.5, 1.0] |

| MNIST-360 | | |
|---|---|---|
| *Method* | *Par* | *Values* |
| SGD | $lr$ | [0.1, 0.2] |
| | $bs$ | [1, 4, 8, 16] |
| ER | $mbs$ | [16, 64, 128] |
| | $lr$ | [0.1, 0.2] |
| | $bs$ | [1, 4, 8, 16] |
| MER | $mbs$ | [128] |
| | $lr$ | [0.1, 0.2] |
| | $\beta$ | [1.0] |
| | $\gamma$ | [1.0] |
| | $nb$ | [1] |
| A-GEM-R | $mbs$ | [16, 64, 128] |
| | $lr$ | [0.1, 0.2] |
| | $bs$ | [1, 4, 16] |
| GSS | $mbs$ | [16, 64, 128] |
| | $lr$ | [0.1, 0.2] |
| | $bs$ | [1, 4, 8, 16] |
| | $gmbs$ | [16, 64, 128] |
| | $nb$ | [1] |
| DER | $mbs$ | [16, 64, 128] |
| | $lr$ | [0.1, 0.2] |
| | $bs$ | [1, 4, 8, 16] |
| | $\alpha$ | [0.5, 1.0] |
| DER++ | $mbs$ | [16, 64, 128] |
| | $lr$ | [0.1, 0.2] |
| | $bs$ | [1, 4, 8, 16] |
| | $\alpha$ | [0.2, 0.5] |
| | $\beta$ | [0.5, 1.0] |

Table 10: Hyperparameter space for Grid-Search