[Reviews · NeurIPS 2020]

Review 1

Summary and Contributions: This paper introduces a method, named dark experience replay (DER), for mitigating catastrophic forgetting that involves maintaining a replay buffer of previous inputs with corresponding past predictions of the network (as opposed to using ground truth labels) and regularising the network’s current predicted logits on stored examples to be close to where they were previously in Euclidean space. DER does not require knowledge of task boundaries to work and uses reservoir sampling for the replay database; a slightly modified version is also proposed, DER++, which additionally stores the ground truth logits and replays examples in the standard way. The methods are evaluated on a number of standard continual learning and incremental learning benchmarks, as well as on a custom protocol (MNIST-360), which combines discrete and continuous changes to the data distribution over time. DER and DER++ are shown to outperform several existing methods on these tasks. Additionally, DER is shown to converge to flatter minima than other methods (which is hypothesised to play a role in its outperformance) and to result in more calibrated networks.

Strengths: - DER and DER++ are very simple methods to implement, and perform impressively well on standard benchmarks. They can serve the continual learning community well as strong, simple baselines. They are computationally efficient and require a fixed amount of memory. - The experimental evaluations are conducted using several existing methods and there appears to have been a thorough effort to optimise them fairly. - An informative analysis of the method is conducted, yielding insights on the flatness of minima and calibration of the model after training.

Weaknesses: - One concern is that DER is quite similar to an existing method [1], which also regularises the functional output of the network to be close to where it was previously using a replay buffer. The differences are that it only stores the outputs at the end of training on each task, and that it compares the softmax outputs rather than the unnormalised logits. In fairness, a comparison with this method is made in the paper, both descriptively and in experiments, and FDR is not directly applicable in a task-agnostic setting. - It is somewhat misleading to display results for the class-incremental setting (where there is no info provided for when a new class/task is introduced) for methods that are not designed to deal with them at all since they work with task boundaries, such as oEWC, SI, LwF and FDR. E.g., since FDR stores outputs after the end of each task, then how does it differ to SGD when only one task is trained on? [1] Benjamin, Ari, David Rolnick, and Konrad Kording. "Measuring and regularizing networks in function space." International Conference on Learning Representations. 2018.

Correctness: Yes, though as described in the weaknesses, it is misleading to provide results for the class-incremental experiments for methods that are not applicable in this setting.

Clarity: Yes, the approach and its evaluation are clearly motivated and described in the paper.

Relation to Prior Work: Yes. As discussed in the weaknesses section, it could be argued that the work is incremental to FDR [1], but the differences are clearly stated in the paper and DER performs better overall in the experimental evaluations. [1] Benjamin, Ari, David Rolnick, and Konrad Kording. "Measuring and regularizing networks in function space." International Conference on Learning Representations. 2018.

Reproducibility: Yes

Additional Feedback: **** POST-REBUTTAL COMMENTS **** Thank you to the authors for their response. I appreciate the clarification on the Class-IL protocol, but I really think it should be made explicit that FDR, EWC, SI, LwF are equivalent to SGD in this setting. While *technically* the application of these methods is correct, it is very misleading because if you are not treating classes as tasks then these methods are simply not applicable and are a meaningless comparison. ***************** - Why are the results for Meta Experience replay shown only for the MNIST360 experiments and not the others? - It is shown empirically that DER converges to flat minima - are there any intuitions as to why this happens?


Review 2

Summary and Contributions: In my understanding, the authors propose a new straightforward approach for incremental learning for image classification with experience replay: Dark Experience Replay (DAR), and an extension called DAR++. The main novelty in the approach is storing both images and corresponding logits in memory based on reservoir sampling. In DER the (stored_image, stored_logit) pairs sampled from memory represent a recording of the model output at different points during incremental training, and are used to introduce an additional loss term (similar to distillation loss), minimizing the L2 loss: L2(stored_logit, current_model(stored_image)). In DER++ an additional sampling from memory is done and a loss term is added where the labels for the stored images are used for classification. Extensive evaluation is done on task-based (Tiny ImageNet, CIFAR10, MNIST) incremental settings and a proposed task-free MNIST360 setting. The proposed method outperforms the majority of prior algorithms.

Strengths: - The DER and DER++ approaches to tackle forgetting make sense and the empirically outperform basic experience replay (storing a buffer of old examples and directly training). - Extensive evaluation against well chosen baseline algorithms - Well motivated analysis (Section 5) comparing the advantages of the proposed method to baselines in addition to accuracy. - The authors include a codebase with implementations of the baseline and proposed methods which looks well implemented. (Was able to understand implementation details of interest quickly, good variable names, comments and docstrings). Significant details w.r.t design decisions and hyperparameter tuning are given in the supplementary materials. - The paper is well written other than the exposition of the proposed method in 3.

Weaknesses: - the authors aim for general continual learning (GCL), but provide results GCL results only on the MNIST 360 task. The MNIST 360 task is good since it allows for abrupt and smooth changes in the data distribution. It would be beneficial to see the performance of DER/DER++ on standard datasets but in a task free setting (no known task boundaries -> no ability to do multiple epochs per task). The paper already has loads of results and provides insight without this particular evaluation, but including this would give a more complete idea of the capabilities of experience replay methods. My other concerns with are outlined in the clarity and relation to prior work sections.

Correctness: I think the motivations/claims about the effectiveness of DER and DER++ are sound and well motivated.

Clarity: I have some issues with the clarity in Section 3. In eq. 2 the KL divergence is applied between current model outputs, and model outputs for all previous tasks. In line 97 network logits become z = h_\theta_t which makes sense for equation (3). The confusion comes when we go to equation (4) where there is z but no description of the time at which these network outputs are recorded. Are the logits stored when the exemplars are stored? Or are the exemplar images from the buffer sent through the model before each parameter update, and then these logits are used for the DER loss term? The z <- h_\theta(x_t) in Algorithm 1 and Algorithm 2 do not help clarify this. Looking at the code, it becomes clear that the memory buffer can store (image, logit, label) triplets and that the image/logit pair is used (in my understanding of the implementation). This is currently not explicitly clear in Section 3, and I think that describing it explicitly will make section 3

Relation to Prior Work: - my main concern is the lack of reference to Chaudhry et al. (On Tiny Episodic Memories in Continual Learning, ICML 2019 workshop paper) which discusses the use of experience-replay based methods as baselines for continual learning and the effect of memory size. The proposed method in Chaudhry et al. and DER/DER++ are different but it would be very useful to see a direct comparison with the findings in Chaudhry et al since there seems to be significant overlap.

Reproducibility: Yes

Additional Feedback: What steps were taken to validate the correctness of the implementations of the algorithms DER and DER++ are compared with? ##### post rebuttal feedback ##### The authors successfully address my concerns regarding the GCL evaluation on datasets beside MNIST and the comparison Tiny Episodic Memories. I will keep the marginal accept rating since the proposed approach is of moderate novelty (it is a strong baseline consisting of reasonable and straightforward steps to adapt SGD based learning to a continual setting). The authors claim that their implementation of the comparisons is correct and validated. I believe that there is value in the publication of this baseline method, since the evaluation is extensive and the code base contains reference implementations of a variety of continual learning methods.


Review 3

Summary and Contributions: This paper proposes a method combining experience replay and knowledge distillation for General Continual Learning (GCL), which does not require task boundaries during training and task identifiers during testing. It proposes a benchmark task MNIST-360 for GCL, which can be beneficial to the community. It also provides empirical study showing correlations among the flatness of local minima, calibration, and the performance of continual learning.

Strengths: The proposed method combines experience replay and functional distance regularisation so that it can take the advantages of knowledge distillation and episodic memory at the same time. It is an efficient approach and the authors also conducted comprehensive experiments. The empirical study about the effects of the flatness of local minima and calibration are interesting, showing the connection between generalization and continual learning. And the proposed task MNIST-360 could be a better benchmark than permuted MNIST.

Weaknesses: The main novelty of the proposed method is just from DER++, where the loss of experience replay has been adapted into the objective. Although the authors claim that FDR is more limited, the objective of DER is very similar as FDR ( section 3.1 of [1]). The difference is mainly pointing to the experimental setting of the applications. The authors seem mix one method with its application setting and made some incorrect claims which will be specified below as well. [1]. Ari S. Benjamin, David Rolnick, and Konrad P. Kording. Measuring and regularizing networks in function space. International Conference on Learning Representations, 2019.

Correctness: In Table 1 the authors claim that GEM, A-GEM, FDR need to know task boundaries during training, it is not true. The methods themselves do not require the knowledge of task boundaries in their objectives or training processes. They can be applied into different settings, such as various memory strategies, like the authors did for A-GEM with reservoir buffer. Changing the memory strategy or using output of different layers does not mean inventing a new method. The reference of ER and MER in tables are the same, according to the text it should be [34] for ER.

Clarity: There are some issues that need to be clarified: 1. How to get Eq. 5 from 4? KL divergence and Euclidean distance are two different things. 2. How to sample (x'', y'') in DER++? If it's as the same as sampling (x',y'), how to guarantee the sharp changes in the data stream being handled? 3. The experiments with CIFAR10 and Tiny ImageNet have multiple epochs. How can this be done without knowing task boundaries? If it cannot, these benchmarks should be tested with 1 epoch to show that the proposed method can actually work not only on MNIST in GCL. 4. How can A-GEM use the same training time as ER (Fig. 2f) while A-GEM needs extra time for computing projected gradient at each iteration? As reported in [2], A-GEM costed about twice time comparing with ER. [2]. Chaudhry, Arslan, et al. "On tiny episodic memories in continual learning." arXiv preprint arXiv:1902.10486 (2019).

Relation to Prior Work: Please refer to above comments. This paper can be a good empirical study based on FDR if the authors could correct their claims and bring deeper insights.

Reproducibility: Yes

Additional Feedback: ##########post author response########## I acknowledge the significance of DER after reading other reviewers comments and the authors' response. However, regarding my comments about the correctness, I'm not convinced. FDR basically is in the same form as DER, when to store the logits is simply a choice of the experimental setting. And for GEM, one QP constraint can be computed by a batch of samples from the memory as the same as computing the reference gradient in A-GEM. I hope authors would correct their claims if this paper got accepted. This paper can be a good empirical study without over-claiming on such issues.


Review 4

Summary and Contributions: The paper provides a strong baseline for general continual learning, where models must learn incrementally from new tasks, classes, or domains, without knowing the boundaries between the different settings a priori. The baseline outperforms prior work, and the authors provide a thorough evaluation of a large number of prior baselines, which may be useful for future research.

Strengths: - The proposed baseline is simple and efficient, while outperforming prior methods on various benchmarks. - The evaluation in Table 2 appears thorough, and is likely to be useful for future research, particularly if code is released for all these methods, as mentioned in the footnote on Page 5. - The MNIST-360 setting, while simple, addresses a major potential concern with the GCL setting: that task boundaries can be easily identified, and the problem may reduce to standard continual learning. - The authors motivate the problem well, and generally provide a clear analysis of the state of current approaches in the field with a simple baseline, which will provide a strong foundation for future research in this direction – assuming all the baselines are correctly implemented (see weaknesses).

Weaknesses: I have a few major concerns with this paper, which I hope the authors can address in the rebuttal: - It is difficult to compare the results in Table 2 with results from the original papers. For example, I tried to compare the presented results for iCARL to the original paper, and could not match the results. For example, my understanding is that the CIFAR-10 Class IL setting should correspond to Fig. 2 (a) top left in iCARL, but the average of the accuracies from that plot does not seem close to 49.02, as reported in the current paper. I think it would be extremely useful to report, where possible, the “original” accuracy as reported in the original paper, and the “reproduced” accuracy, to more clearly show which evaluations are new, and which evaluations are reproduced. - Relatedly, it’s not possible to evaluate whether the authors correctly reproduced prior work in their implementation based on the paper. I believe sharing both the original and reproduced accuracies as mentioned above would help, but it would also be useful to address the general protocol used, and mark any methods for which the original numbers could not be reproduced with an asterisk. - Finally, while DER is simple, it’s relation to prior work needs to be made more clear. Which components are novel, and which are used in prior work? To clarify: even if every component is used in prior work, I think DER would still be a useful baseline! I am not concerned that DER is made up of existing components, but rather that it is unclear to the reader which components are new, and which are not.

Correctness: Yes, but it is unclear if the reimplemented methods match the original results.

Clarity: yes

Relation to Prior Work: Can be discussed more clearly; see weaknesses.

Reproducibility: Yes

Additional Feedback: In Sec 3, it’s highly unclear which components are specific to DER, vs. which can be found in prior work. For example, L96 states that DER is different from prior work in that it stores network logits; does this mean that no prior work stores network logits? This seems unlikely; e.g., iCARL seems to also have a distillation loss which stores the logits (see Algorithm 3). Similarly, is reservoir sampling used by prior work? # Post-rebuttal comments Thanks for your detailed responses to the main concerns. I again request that for each method, where possible, the authors report both the accuracy reported in the original paper introducing the method, and the accuracy from the new implementation. I appreciate the implementation differences in L26-29 in the response, and request that the authors similarly detail any other key differences for all methods in the supplement. I understand this is time-consuming, but I believe it is critical: Given the wide variety of methods evaluated, this paper may often be referenced when comparing existing methods, and the community needs to be able to trust the results. Additionally, I encourage the authors to carefully respond to R1 and R3's concern regarding how FDR, GEM and A-GEM are represented. More generally, I still feel that it is unclear which components of DER/DER++ are new, and which are not, as mentioned in my original review. It is challenging to fit all this in 8 pages, but it would be useful to at least expand on these topics in the supplement. Finally, the reviewers have raised a number of detailed questions. While it is impossible to address these in the 1 page response, I trust that the authors will address these responses in their final paper or supplement.

[Author Response · NeurIPS 2020]



Figure A: Comparison with Fig. 1 of *Chaudhry et al.*

Table A: Single-epoch evaluation setting (Class-IL).

| | Buffer | ER | FDR | DER++ | JOINT | JOINT |
|---|---|---|---|---|---|---|
| **#epochs** | | 1 | 1 | 1 | 1 | 50/100 |
| Seq. | 200 | 37.64 | 21.22 | **41.93** | | |
| CIFAR | 500 | 45.22 | 21.06 | **48.04** | 56.74 | 92.20 |
| 10 | 5120 | 50.28 | 20.57 | **53.31** | | |
| Seq. | 200 | 5.98 | 4.87 | **6.35** | | |
| Tiny | 500 | 8.39 | 4.76 | **8.65** | 19.37 | 59.99 |
| ImgNet | 5120 | 16.04 | 4.96 | **16.41** | | |

**R1** *[Misleading comparisons]* We were probably not clear about the Class-IL protocol (we will clarify Sec. 4.1): if a
method needs task boundaries (*e.g.* oEWC or LwF), we always provide them during training. Hence, we consider our
comparison fair and in line with other works [39, 32, 41, 11]. Moreover, FDR regularizes from the second task onwards,
which makes it indistinguishable from SGD during the first one — like EWC, SI, LwF, GEM and others.

**R2** *[On Tiny Episodic Memories in Continual Learning]* We already draw a thorough comparison with ER-Reservoir,
which, according to Chaudry et al., "*works the best across the board* [among ER baselines] *except when the memory size*
*is very small*". However, we here provide a comparison with the experiments of Chaudhry et al. (Fig. A), showing that
DER++, despite relying on reservoir, outperforms all ER-based methods even for the smallest memory sizes provided.

**R2, R3** *[Number of epochs]* We conceive a task as a mere sequence of batches drawn from the dataset: in the case
of multiple epochs, this sequence simply results longer to the model. In doing so, our implementation decouples the
length of the task from the notion of task boundary: for the latter, an oracle notifies the model about the end of the task
through a *callback*. As an example, EWC exploits this callback to compute the Fisher Information Matrix; instead, our
method ignores this call as it does not need to take any action at boundaries. Under this perspective, the chance of doing
multiple epochs results orthogonal to the use of task boundaries.
Although the hint about the investigation of a single-epoch setup is compelling, we believe that it could be problematic
for difficult datasets such as CIFAR-10 and TinyImageNet. If we limit ourselves to a single pass (few gradient steps),
we struggle to disentangle the effects of catastrophic forgetting (the focus of our work) from those of underfitting. The
single-epoch experiment asked by the reviewers (Tab. A) reveals indeed that even the joint training (upper bound)
yields a dramatically low accuracy w.r.t. multi-epoch (see last two columns). Among CL methods, DER++ confirms its
reliability approaching the single-epoch joint training. Nevertheless, we feel that future GCL works should be conscious
of the above-mentioned points when dealing with single-epoch split setups (we will add these considerations in the
appendix). Our MNIST-360 (the first to our knowledge to match the requirements of GCL [10]) points in this direction.

**R2, R4** *[Implementation]* Whenever available, we referred to our competitors' official repositories; additionally, we
validated our implementations by matching the results of the original papers in their specific experimental settings.
While R2 is satisfied with our codebase after having reviewed it, the doubts of R4 are due to a mismatch between the
performance of our iCaRL implementation and the results presented in the original paper [32]. We ascribe this to some
differences in the experimental protocol: i) Rebuffi et al. tested on CIFAR-100, whereas we used CIFAR-10; ii) they
relied on ResNet-32, we used ResNet-18; iii) R4 considers the average of the accuracies shown in Fig. 2 (a-top left), but
only the last point ($\approx 40\%$) should be considered as our results are expressed as final average accuracy.

**R3** *[Novelty]* Although we acknowledge that DER and FDR are similar in their objective, our work highlights how
an apparently subtle difference – storing logits throughout the optimization trajectory – substantially changes the CL
training regime. As appreciated by R1 and R2, our experiments show that this strategy dramatically improves over FDR
(with a peak of $+64.11$ and $+7.76$ of accuracy on S-CIFAR10 and S-TinyImg, see Tab. 2) and delivers more remarkable
properties (Sec. 5). We consider this finding novel and interesting for the community; we believe this will foster new
research and advances regarding rehearsal methods and the use of trajectory information for distilling knowledge.

**R3** *[The need for task boundaries]* While we agree about A-GEM (we will fix Tab. 1), FDR specifies that responses
are stored at the end of the task, so it needs boundaries to store them. Our position on GEM comes from its need of
examples labeled with task ids, as it demands one QP-constraint per task. Task ids assume the presence of boundaries
(and *viceversa*), even though we agree that GEM does not take specific actions at boundaries. If R3 thinks it would be
more correct, we will mark GEM as free from task boundaries and be more specific about its dependence on task ids.

**R3** *[Eq. 5]* Hinton et al. provide a full derivation in Sec. 2.1 of [16] stating that, under mild assumptions, the derivative
of $D_{KL}$ between post-softmax outputs approaches the one of MSE between logits. We will make this passage clearer.

**R3** *[Sampling in DER++]* It is not about the sampling strategy; DER++ handles sharp distribution changes by storing
labels $y$, whose training signal is more reliable than the one provided by logits $z$ stored at the beginning of a new task.

**R3** *[Training time]* A-GEM's training time is comparable with ER's as we apply data augmentation on buffer examples
for ER (and not for A-GEM, for which is detrimental, see Footnote 2). Otherwise, ER trains 1.7x faster than A-GEM.

[Meta-Review · NeurIPS 2020]

This paper proposes a simple yet highly effective method for continual learning that performs extremely well. It is well written, thorough, novel, timely, and interesting, with the potential for broad impact on the subfield of continual learning. However, there were concerns among multiple reviewers about how the paper represents prior work (specifically how FDR, GEM and A-GEM are represented by the authors), and how DER/DER++ are different from existing methods. Given the potential of the impact of this paper, the authors have a responsibility to correct these characterizations of prior work, and it is expected for the authors to do so before publication.